# Generalizable, real-time neural decoding with hybrid state-space models

**Avery Hee-Woon Ryoo**[1,2,*,✉]    **Nanda H Krishna**[1,2,*,✉]    **Ximeng Mao**[1,2,*,✉]
**Mehdi Azabou**[3]    **Eva L Dyer**[4]    **Matthew G Perich**[1,2,†]    **Guillaume Lajoie**[1,2,5,†,✉]
[1]Mila – Quebec AI Institute    [2]Université de Montréal    [3]Columbia University
[4]University of Pennsylvania    [5]Canada CIFAR AI Chair    [*]Co-first authors    [†]Co-senior authors
[✉]{hee-woon.ryoo,nanda.harishankar-krishna,ximeng.mao,guillaume.lajoie}@mila.quebec
https://possm-brain.github.io

## Abstract

Real-time decoding of neural activity is central to neuroscience and neurotechnology applications, from closed-loop experiments to brain-computer interfaces, where models are subject to strict latency constraints. Traditional methods, including simple recurrent neural networks, are fast and lightweight but often struggle to generalize to unseen data. In contrast, recent Transformer-based approaches leverage large-scale pretraining for strong generalization performance, but typically have much larger computational requirements and are not always suitable for low-resource or real-time settings. To address these shortcomings, we present POSSM, a novel hybrid architecture that combines individual spike tokenization via a cross-attention module with a recurrent state-space model (SSM) backbone to enable (1) fast and causal online prediction on neural activity and (2) efficient generalization to new sessions, individuals, and tasks through multi-dataset pretraining. We evaluate POSSM's decoding performance and inference speed on intracortical decoding of monkey motor tasks, and show that it extends to clinical applications, namely handwriting and speech decoding in human subjects. Notably, we demonstrate that pretraining on monkey motor-cortical recordings improves decoding performance on the human handwriting task, highlighting the exciting potential for cross-species transfer. In all of these tasks, we find that POSSM achieves decoding accuracy comparable to state-of-the-art Transformers, at a fraction of the inference cost (up to $9\times$ faster on GPU). These results suggest that hybrid SSMs are a promising approach to bridging the gap between accuracy, inference speed, and generalization when training neural decoders for real-time, closed-loop applications.

## 1 Introduction

Neural decoding – the process of mapping neural activity to behavioural or cognitive variables – is a core component of modern neuroscience and neurotechnology. As neural recording techniques evolve and datasets grow in size, there is increasing interest in building generalist decoders that scale and flexibly adapt across subjects and experiments. Several important downstream applications – including closed-loop neuroscience experiments and brain-computer interfaces (BCIs) – require fine-grained, low-latency decoding for real-time control [1]. Advances in these technologies would enable next-generation clinical interventions in motor decoding [2], speech prostheses [3], and closed-loop neuromodulation [4, 5]. Building towards these applications will require neural decoders that meet three requirements: (1) robust and accurate predictions, (2) causal, low-latency inference that is viable in an online setting, and (3) flexible generalization to new subjects, tasks, and experimental settings. Although recent developments in machine learning (ML) have enabled significant strides in each of these axes, building a neural decoder that achieves all three remains an open challenge.

39th Conference on Neural Information Processing Systems (NeurIPS 2025).

Recurrent neural networks (RNNs) [6] and attention-based models such as Transformers [7] have shown significant promise for neural decoding tasks [8–10]. RNNs (Figure 1a) offer fast, low-latency inference on sequential data and strong performance when trained on specific tasks [10]. However, their ability to generalize to new subjects is limited due to their rigid input format. Specifically, their reliance on fixed-size, time-binned inputs means that they typically cannot learn from new sessions with different neuron identities or sampling rates without full re-training and/or modifying the architecture. In contrast, Transformer-

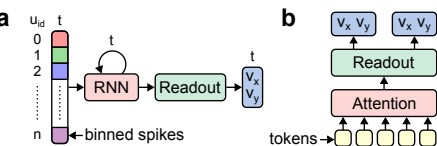

Figure 1: **Existing deep learning models for neural decoding.** (a) Recurrent neural networks (RNNs). (b) Attention-based models such as Transformers.

based architectures (Figure 1b) offer greater flexibility thanks to more adaptable neural tokenization approaches [11–13]. Nonetheless, they struggle with applications involving real-time processing due to their quadratic computational complexity, in addition to the overall computational load of the attention mechanism. Recent efforts in sequence modelling with large language models have explored hybrid architectures that combine recurrent layers, such as gated recurrent units (GRUs) [14] or state-space models (SSMs) [15–18], with attention layers. These models show encouraging gains in long-context understanding and computational efficiency [19–21]. While hybrid attention-SSM approaches offer a promising solution for real-time neural decoding, to the best of our knowledge, they remain unexplored in this area.

We address this gap with POSSM[1], a hybrid model that combines the flexible input-processing of Transformers [22] with the efficient, online inference capabilities of a recurrent SSM backbone. Unlike traditional methods that rely on rigid time-binning, POSSM operates at a millisecond-level resolution by tokenizing individual spikes. In essence, POSSM builds on a POYO-style cross-attention encoder [13] that projects a variable number of spike tokens to a fixed-size latent space. The resulting output is then fed to an SSM that updates its hidden state across consecutive chunks in time. This architecture, as illustrated in Figure 2, offers two key benefits: (1) the recurrent backbone allows for lightweight, constant-time predictions over consecutive chunks of time and (2) by adopting POYO's spike tokenization, encoding, and decoding schemes, POSSM can effectively generalize to different sessions, tasks, and subjects.

In this paper, we introduce the POSSM architecture and evaluate its performance on intracortical recordings of spiking activity from experiments in both non-human primates (NHPs) and humans. Although our current evaluations are conducted offline, POSSM is designed for real-time inference and can be readily implemented for online experiments. Finally, while this paper is concerned with invasive electrophysiology recordings, this method could be extended further to other modalities using POYO-style tokenization (see Section 5 for a discussion). Our contributions are summarized as follows:

- **Performance and efficiency:** We evaluate POSSM against other popular models using NHP datasets that contain multiple sessions, subjects, and different reaching tasks (centre-out, random target, maze, etc.). We find that POSSM matches or outperforms other models on all these tests, doing so with greater speed and significantly reduced computational cost.

- **Multi-dataset pretraining improves performance:** Through large-scale pretraining, we find that POSSM delivers improved performance on NHP datasets across sessions, subjects, and even tasks.

- **Cross-species transfer learning:** Pretraining on diverse NHP datasets and then finetuning POSSM leads to state-of-the-art performance when decoding imagined handwritten letters from human cortical activity [23]. This cross-species transfer not only outlines the remarkable transferability of neural dynamics across different primates, but also shows the potential for leveraging abundant non-human data to augment limited human datasets for improved decoding performance.

- **Long sequence complex decoding:** When trained on human motor-cortical data during attempted speech [3], POSSM achieves strong decoding performance. In contrast, attention-based models struggle with the task's long-context demands, making them computationally prohibitive.

---

[1] POSSM stands for POYO-SSM, and is pronounced like the animal ("possum"; IPA: /ˈpɒsəm/).

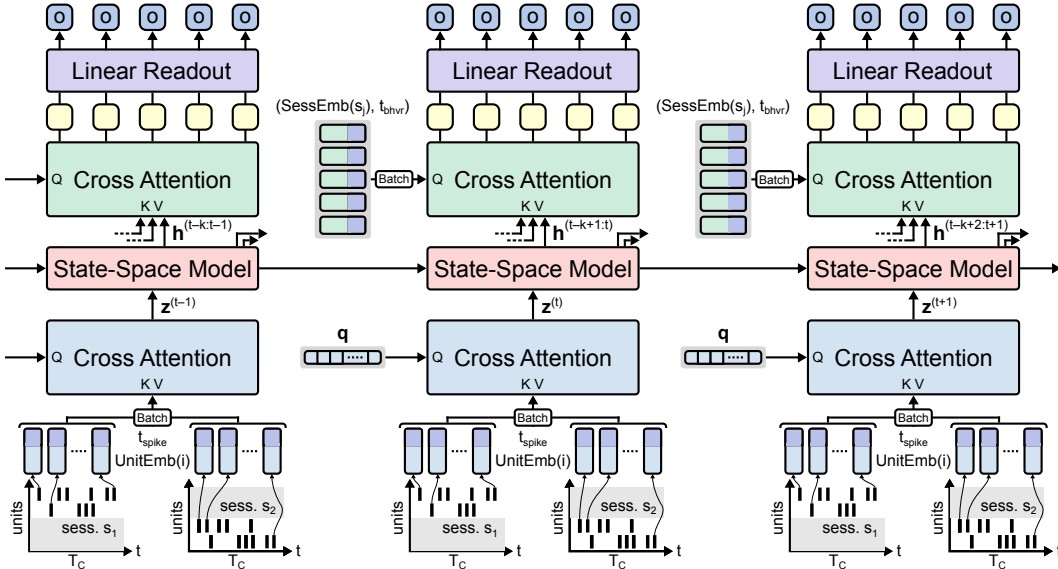

Figure 2: **An architecture for generalizable, real-time neural decoding.** POSSM combines individual spike tokenization [12] and input-output cross-attention [22] with a recurrent SSM backbone. In this paper, we typically consider $k = 3$ and $T_c = 50$ ms.

## 2 Methods

We focus on decoding neuronal spiking activity, which consists of discrete events triggered when excitatory input to a neuron exceeds a certain threshold. The timing and frequency of spikes encode information conveyed to other neurons throughout the brain, underlying a communication system that is central to all neural function [24, 25]. However, their sparse nature [26] requires an effective input representation that can handle their temporal irregularity. Furthermore, there exists no direct mapping between the neurons of one living organism to another, highlighting the non-triviality of the alignment problem when training across multiple individuals. In this section, we describe how POSSM is designed to sequentially process sub-second windows of these spikes for the online prediction of behaviour, while maintaining a flexible input framework that allows it to efficiently generalize to an entirely new set of neurons during finetuning.

### 2.1 Input processing

**Streaming neural activity as input.** Our focus is on real-time performance, which requires minimal latency between the moment that neural activity is observed and when a corresponding prediction is generated. This constraint significantly limits the time duration, and consequently the amount of data, that a model can use for each new prediction. To this end, POSSM maintains a hidden state as data is streamed in, allowing it to incorporate past information without reprocessing previous inputs. In each chunk, the number of spikes varies, meaning each input is represented as a variable-length sequence of spikes. While POSSM generally uses contiguous 50 ms time chunks here, we also demonstrate strong performance with 20 ms chunks (see Section D.1). In theory, these windows could even be shorter or longer (and even overlapping) depending on the task, with the understanding that there would be some trade-off between temporal resolution and computational complexity.

**Spike tokenization.** We adopt the tokenization scheme from the original POYO model, where each neuronal spike is represented using two pieces of information: (1) the identity of the neural unit which it came from and (2) the timestamp at which it occurred (see Figure 2). The former corresponds to a unique learnable unit embedding for each neuron, while the latter is encoded with a rotary position embedding (RoPE) [27] that allows the tokens to be processed based on their relative timing rather than their absolute timestamps. For example, a spike from some neuron with integer ID $i$ that occurs

at time $t_{\text{spike}}$ would be represented as a $D$-dimensional token $\mathbf{x}$, given by:

$$\mathbf{x} = (\text{UnitEmb}(i), t_{\text{spike}}),$$

where $\text{UnitEmb} : \mathbb{Z} \to \mathbb{R}^D$ is the unit embedding map and $D$ is a hyperparameter. As tokenization is an element-wise operation, a time chunk with $N$ spikes will yield $N$ spike tokens. We opt to use the general term "neural unit", as spikes could be assigned at a coarser specificity than an individual neuron (e.g., multi-unit activity on a single electrode channel) depending on the dataset and task at hand. This flexibility, coupled with the model's ability to handle a variable number of units, facilitates both training and efficient finetuning (see Section 2.3) on multiple sessions and even across datasets.

## 2.2  Architecture

**Input cross-attention.**  We employ the original POYO [12] encoder, where a cross-attention module inspired by the PerceiverIO architecture [22] compresses a variable-length sequence of input spike tokens into a fixed-size latent representation. Unlike POYO, however, the encoder is applied on short 50 ms time chunks, each of which is mapped to a single latent vector. This is achieved by setting the spike tokens as the attention keys and values, and using a learnable query vector $\mathbf{q} \in \mathbb{R}^{1 \times M}$, where $M$ is a hyperparameter. Given a sequence $\mathbf{X}_t = [\mathbf{x}_0, \mathbf{x}_1, ..., \mathbf{x}_N]^\top \in \mathbb{R}^{N \times D}$ of $N$ spike tokens from some chunk of time indexed by $t$, the latent output of the cross-attention module is computed as such:

$$\mathbf{z}^{(t)} = \text{softmax}\left(\frac{\mathbf{q}\mathbf{K}_t^\top}{\sqrt{D}}\right)\mathbf{V}_t,$$

where $\mathbf{K}_t = \mathbf{X}_t \mathbf{W}_k$ and $\mathbf{V}_t = \mathbf{X}_t \mathbf{W}_v$, with $\mathbf{W}_k, \mathbf{W}_v \in \mathbb{R}^{D \times M}$, are the projections of the input token sequence, following standard Transformer notation. Following POYO, our implementation also uses the standard Transformer block with pre-normalization layers and feed-forward networks.

**Recurrent backbone.**  The output of the cross-attention $\mathbf{z}^{(t)}$ is then fed to an SSM (or another variety of RNN), which we refer to as the recurrent backbone. The hidden state of the recurrent backbone is updated as follows:

$$\mathbf{h}^{(t)} = f_{\text{SSM}}(\mathbf{z}^{(t)}, \mathbf{h}^{(t-1)}).$$

While the input cross-attention captures local temporal structure (i.e., within the 50 ms chunk), the SSM integrates this information with historical context through its hidden state, allowing POSSM to process information at both local and global timescales. We run experiments with three different backbone architectures: diagonal structured state-space models (S4D) [15], GRU [14], and Mamba [16]. However, we wish to note that this method is compatible with any other type of recurrent model. Specifics regarding each of these backbone architectures can be found in Section B.2.

**Output cross-attention and readout.**  To decode behaviour, we select the $k$ most recent hidden states $\{\mathbf{h}^{(t-k+1):(t)}\}$ ($k = 3$ in our experiments), and use a cross-attention module to query them for behavioural predictions. For a given time chunk $t$, we generate $P$ queries, one for each timestamp at which we wish to predict behaviour. Each query encodes the associated timestamp (using RoPE), along with a learnable session embedding that captures latent factors of the recording session. The design of our output module enables flexibility in several ways: (1) we can predict multiple outputs per time chunk, enabling a denser and richer supervision signal, (2) we are not required to precisely align behaviour to chunk boundaries, and (3) we can predict behaviour beyond the current chunk, allowing us to account for lags between neural activity and behavioural action (see Section D.5).

## 2.3  Generalizing to unseen data with pretrained models

Given a pretrained model, we outline two strategies for adapting it to a new set of neural units, with a trade-off between efficiency and decoding accuracy.

**Unit identification.**  To enable efficient generalization to previously unseen neural units, we adopt *unit identification (UI)*, a finetuning strategy enabled by the spike tokenization scheme adopted from POYO [12]. In UI, new neural units can be processed by a model simply by learning new embeddings. Using this approach, we freeze the model weights, initialize new unit and session embeddings, and then train them on the data from the new session while the rest of the model is kept unchanged. This allows us to preserve the neural dynamics learned during pretraining, resulting in an efficient generalization strategy that typically updates less than 1% of the model's total parameters.

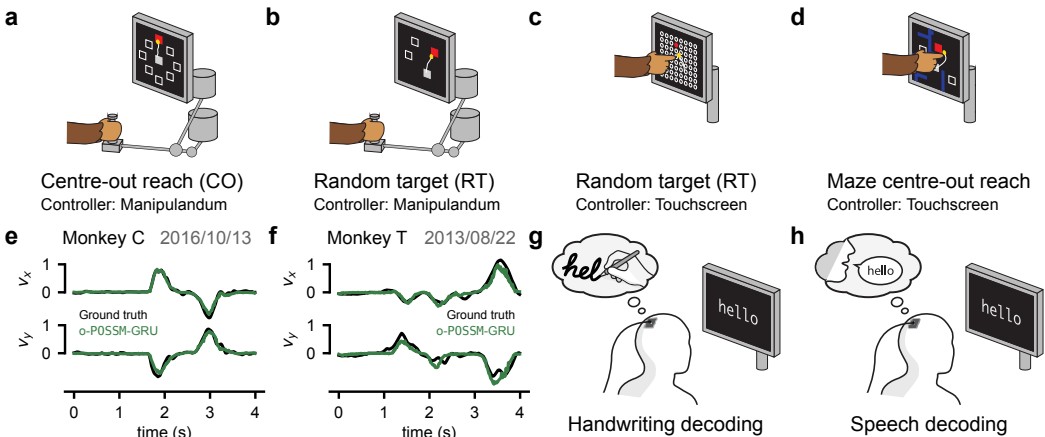

Figure 3: **Task schematics and outputs.** (a) Centre-out (CO) task with a manipulandum. (b) Random target (RT) task with a manipulandum. (c) RT task with a touchscreen. (d) Maze task with a touchscreen. (e) Ground truth vs. predicted behaviour outputs from a held-out CO session. (f) Same as (e) but for an RT session. (g) Human handwriting decoding task. (h) Human speech decoding task.

**Full finetuning.** While unit identification is an efficient finetuning method, it does not reliably match the performance of models trained end-to-end on individual sessions. To address this, we also explored *full finetuning (FT)*, a complementary approach which uses a gradual unfreezing strategy. We begin by only doing UI for some number of epochs before unfreezing the entire model and training end-to-end. This allows us to retain the benefits of pretraining while gradually adapting to the new session. As shown below, full finetuning consistently outperforms both single-session training and unit identification across all tasks explored, demonstrating effective transfer to new animals, tasks, and remarkably, across species.

## 3 Experiments

We evaluate POSSM across three categories of cortical activity datasets: NHP reaching tasks, human imagined handwriting, and human attempted speech. For the NHP reaching tasks, we highlight the benefits of scale by introducing o-POSSM, a POSSM variant pretrained on multiple datasets. We compare it to single-session models on held-out sessions with varying similarity with the training data, demonstrating improved decoding accuracy with pretraining. Next, we show that o-POSSM achieves powerful downstream decoding performance on a human handwriting task, illustrating the POSSM framework's capacity for cross-species transfer. Finally, we demonstrate that POSSM effectively leverages its recurrent architecture to efficiently decode human speech – a long-context task that can become computationally expensive for standard Transformer-based models with quadratic complexity. In each of these tasks, we see that POSSM consistently matches or outperforms other architectures in a causal evaluation setting, with performance improving as model size and pretraining scale increase.

### 3.1 Non-human primate reaching

We first evaluate POSSM on the task of decoding two-dimensional hand velocities in monkeys performing various reaching tasks (shown in Figure 3a-f). Each training sample consists of 1 s of spiking activity paired with the corresponding 2D hand velocity time series over that same interval. This 1 s window is split up into 20 non-overlapping chunks of 50 ms, which are fed sequentially to POSSM (see Section D.1 for results on 20 ms chunks). This streaming setup enables efficient real-time decoding, where only the most recent 50 ms chunk is processed at each step. In sharp contrast, the original POYO model reprocesses an entire 1 s window of activity with each new input, resulting in significantly higher computational cost.

**Experimental Setup.** We use a similar experimental setup to POYO [12]. The pretraining dataset includes four NHP reaching datasets collected by different laboratories [28–31], covering three types

Table 1: **Behavioural decoding results on NHP reaching tasks.** Values are reported as mean $R^2$ ($\uparrow$) $\pm$ SD over sessions. The number of sessions in each dataset is contained in ($\cdot$). C, T, and H are subject identifiers. SS: Single-session; UI: Unit identification; FT: Full finetuning. The top performing models in each category are indicated in boldface (1st) and underlined (2nd).

| | Method | Same animal, other days | | New animal | | New dataset |
| --- | --- | --- | --- | --- | --- | --- |
| | | C – CO 2016 (2) | C – CO 2010 (5) | T – CO (6) | T – RT (6) | H – CO (1) |
| FROM SCRATCH | MLP | $0.9210 \pm 0.0010$ | $0.6946 \pm 0.0912$ | $0.7976 \pm 0.0220$ | $0.7007 \pm 0.0774$ | 0.4179 |
| | S4D | $0.9381 \pm 0.0083$ | $0.6619 \pm 0.0844$ | $0.8526 \pm 0.0243$ | $0.7145 \pm 0.0671$ | 0.3942 |
| | Mamba | $0.9287 \pm 0.0034$ | $0.6708 \pm 0.1154$ | $0.7692 \pm 0.0235$ | $0.6694 \pm 0.1220$ | 0.7231 |
| | GRU | $0.9376 \pm 0.0036$ | $0.7308 \pm 0.1089$ | $0.8453 \pm 0.0200$ | $0.7279 \pm 0.0679$ | 0.8128 |
| | POYO-SS | $0.9427 \pm 0.0019$ | $0.7381 \pm 0.0887$ | $0.8705 \pm 0.0193$ | $0.7156 \pm 0.0966$ | 0.4974 |
| | POSSM-S4D-SS | $0.9515 \pm 0.0021$ | $0.7768 \pm 0.0993$ | $\underline{0.8838} \pm 0.0171$ | $\underline{0.7505} \pm 0.0735$ | $\underline{0.8163}$ |
| | POSSM-Mamba-SS | $\mathbf{0.9550} \pm 0.0003$ | $\mathbf{0.7959} \pm 0.0772$ | $0.8747 \pm 0.0173$ | $0.7418 \pm 0.0790$ | $\mathbf{0.8350}$ |
| | POSSM-GRU-SS | $\underline{0.9549} \pm 0.0012$ | $\underline{0.7930} \pm 0.1028$ | $\mathbf{0.8863} \pm 0.0222$ | $\mathbf{0.7687} \pm 0.0669$ | 0.8161 |
| PRETRAINED | NDT-2 (FT) | $0.9258 \pm 0.0031$ | $0.7846 \pm 0.1167$ | $0.7173 \pm 0.0443$ | $0.6323 \pm 0.1339$ | 0.8233 |
| | POYO-1 (UI) | $0.9580 \pm 0.0033$ | $0.7883 \pm 0.0846$ | $0.8181 \pm 0.0428$ | $0.7073 \pm 0.0960$ | 0.7861 |
| | POYO-1 (FT) | $0.9580 \pm 0.0038$ | $\mathbf{0.8300} \pm 0.0788$ | $0.8859 \pm 0.0317$ | $\underline{0.7718} \pm 0.0785$ | 0.8601 |
| | o-POSSM-S4D (UI) | $0.9603 \pm 0.0029$ | $0.7935 \pm 0.0924$ | $0.8763 \pm 0.0259$ | $0.7333 \pm 0.0717$ | 0.8394 |
| | o-POSSM-Mamba (UI) | $0.9607 \pm 0.0036$ | $0.7504 \pm 0.0959$ | $0.8729 \pm 0.0178$ | $0.7388 \pm 0.0702$ | 0.7533 |
| | o-POSSM-GRU (UI) | $0.9595 \pm 0.0035$ | $0.8022 \pm 0.0818$ | $0.8921 \pm 0.0174$ | $0.7464 \pm 0.0692$ | 0.8220 |
| | o-POSSM-S4D (FT) | $0.9609 \pm 0.0042$ | $\underline{0.8216} \pm 0.0945$ | $\mathbf{0.9068} \pm 0.0170$ | $0.7605 \pm 0.0619$ | $\mathbf{0.8941}$ |
| | o-POSSM-Mamba (FT) | $\mathbf{0.9615} \pm 0.0056$ | $0.8195 \pm 0.0750$ | $0.9001 \pm 0.0131$ | $0.7612 \pm 0.0722$ | $\underline{0.8838}$ |
| | o-POSSM-GRU (FT) | $\underline{0.9614} \pm 0.0029$ | $0.8161 \pm 0.0912$ | $\underline{0.9024} \pm 0.0159$ | $\mathbf{0.7741} \pm 0.0683$ | 0.8743 |

of reaching tasks: centre-out (CO), random target (RT), and Maze. CO is a highly-structured task involving movements from the centre of the screen to one of eight targets (Figure 3a). Conversely, the RT (Figure 3b-c) and Maze (Figure 3d) tasks are behaviourally more complex, requiring movement to randomly placed targets and navigating through a maze, respectively. The testing sessions include: (1) new sessions from Monkey C which was seen during pretraining, (2) new sessions from Monkey T, not seen during pretraining but collected in the same lab as Monkey C, and (3) a new session from a dataset unseen during pretraining. We call our model pretrained on this dataset o-POSSM (see Section B.3 for details). In total, o-POSSM was trained on 148 sessions comprising more than 670 million spikes from 26,032 neural units recorded across the primary motor (M1), dorsal premotor (PMd), and primary somatosensory (S1) cortices (see Section A for details). We also pretrain two baseline models, NDT-2 [11] and POYO-1 [12]. Additionally, we report the results of single-session models trained from scratch on individual sessions. This includes single-session variants of POSSM (across all backbone architectures) and POYO, as well as other baselines such as a multi-layer perceptron (MLP) [32], S4D [15], GRU [14], and Mamba [16].

**Causal evaluation.** To simulate real-world decoding scenarios, we adopt a causal evaluation strategy for all models. This is straightforward for POSSM and the recurrent baselines we consider – sequences are split into 50 ms time chunks and fed sequentially to the model. For the MLP and POYO, we provide a fixed 1 s history of neural activity at each inference timestep, sliding forward in small increments of 50 ms to collect predictions for all behavioural timestamps. For POYO, we only recorded predictions for timestamps in the final 50 ms of each 1 s context window.

During training, the models are presented with contiguous and non-overlapping 1 s sequences, which are not trial-aligned. However, we evaluate our models on entire trials, which are typically much longer than a second. For example, in sessions from the Perich et al. [28] dataset, trials in the validation and testing sets are at least $3\times$ and up to $5\times$ longer than the training sequences. This means that a recurrent model must generalize to sequences longer than the ones seen during training.

**Transfer to new sessions.** In Table 1, we evaluate the transferability of our pretrained models to new sessions, animals, and datasets, and compare them to single-session models. When trained on a single session, POSSM is on par with or outperforms POYO on most sessions. When using pretrained models, o-POSSM-S4D shows the best overall performance. Finally, we observe that FT is noticeably better than UI when transferring to new animals or datasets. However, UI performs on par with or better than several single-session models, and requires far fewer parameters to be trained.

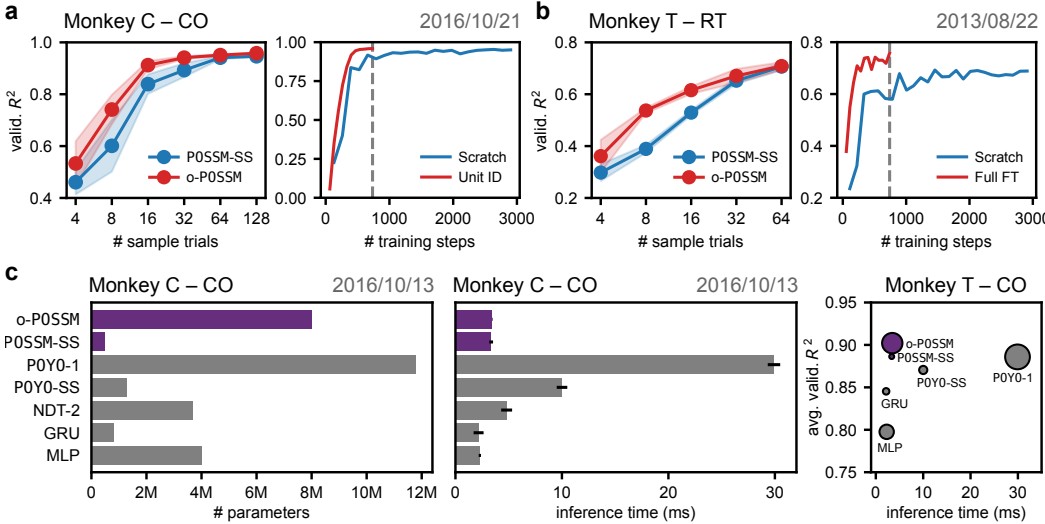

Figure 4: **Sample and compute efficiency benchmarking.** (a) Results on a held-out CO session from Monkey C [28]. On the left, we show the sample efficiency of adapting a pretrained model versus training from scratch. On the right, we compare training compute efficiency between these two approaches. (b) Same as (a) but for a held-out RT session from Monkey T [28] – a new subject not seen during training. (c) Comparing model performance and compute efficiency to baseline models. Inference times are computed on a workstation-class GPU (NVIDIA RTX8000). For all these results, we used a GRU backbone for POSSM.

**Sample and training compute efficiency.**    A key motivation behind training neural decoders on large, heterogeneous datasets is to enable efficient transfer to new individuals, tasks, or species with minimal finetuning. To this end, we evaluated the effectiveness of our finetuning strategies through few-shot experiments. Our results, shown in Figure 4a-b, demonstrate that o-POSSM outperforms single-session models trained from scratch in low-data regimes. Notably, we observe that pretraining results in a considerable initial boost in performance when adapting to a new session, even when only unit and session embeddings are updated. In some cases, single-session models fail to match the performance of a finetuned model, even after extensive training. Overall, our results are in line with observations from Azabou et al. [12], supporting the idea that with the right tokenization and data aggregation schemes, pretrained models can amortize both data and training costs, leading to efficient adaptation to downstream applications.

**Inference efficiency.**    In addition to being efficient to train and finetune, we also evaluated whether POSSM is efficient at inference time. In Figure 4c, we report parameter counts and inference time per time chunk on a workstation-class GPU (NVIDIA RTX8000) for POSSM and several state-of-the-art neural decoders. We find that our single and multi-session models achieve inference speeds that are comparable to lightweight models like GRUs and MLPs and significantly lower than complex models like NDT-2 and POYO, while retaining competitive performance. Single-session POSSM models (POSSM-SS) contained the fewest parameters, and even o-POSSM, with about 8M parameters, maintained low latency. These results held in a CPU environment (AMD EPYC 7502 32-Core) as well, with POSSM-SS and o-POSSM achieving inference speeds of ∼2.44 ms/chunk and ∼5.65 ms/chunk, respectively. Overall, our results show that POSSM's inference time is well within the optimal real-time BCI decoding latency of ≤ 10 ms [8], making it a viable option for real-time BCI decoding applications.

## 3.2 Human handwriting

Next, we evaluated POSSM on a human handwriting dataset [23]. This dataset contains 11 sessions recorded from a single individual, where they imagined writing individual characters and drawing straight lines (Figure 3g). Spike counts from multi-unit threshold crossings were recorded from two 96-channel microelectrode arrays implanted in the participant's motor cortex [23]. These were then

binned at 10 ms intervals. The models were trained to classify the intended characters or lines based on this neural activity. For evaluation, we used 9 sessions, each containing 10 trials per character class. Each individual trial consisted of a 1.6 s time-window centred around the "go" cue.

**Results.** We compared POSSM with five baselines: the previously published statistical method PCA-KNN [23], GRU [14], S4D [15], Mamba [16] and POYO. For POSSM and all baselines except POYO, we adopted the causal evaluation strategy described in Section 3.1, training on 1 s intervals and evaluating on full 1.6 s trials. For POYO, we followed the evaluation scheme from the original paper, using fixed 1 s context windows for both training and testing.

As shown in Table 2, POSSM-GRU outperforms all baseline models when trained from scratch on the 9 sessions. **Remarkably, finetuning o-POSSM, which was only pretrained on NHP data, led to significant performance gains**: 2% for POSSM-GRU and more than 5% for both POSSM-S4D and POSSM-Mamba. All of the POSSM models achieve state-of-the-art performance on this task, with the finetuned o-POSSM variants considerably outperforming the baseline PCA-KNN, achieving test accuracies that are about 16% greater.

These results establish a critical finding: neural dynamics learned from NHP datasets can generalize across different species performing distinct tasks. This is especially impactful given the challenges of collecting large-scale human electrophysiology datasets, suggesting that the abundance of NHP datasets can be used to effectively improve human BCI decoders.

Table 2: **Human handwriting decoding results.** Values are mean accuracy $\pm$ SD for classifying the intended character or line, over 9 sessions and 3 seeds. $*,**$: $p \leq 0.05$, $0.01$ for a paired t-test vs. POSSM-GRU.

| | Method | Acc. (%) ↑ |
|---|---|---|
| FROM SCRATCH | PCA-KNN (SS) | $81.36 \pm 7.53$ |
| | S4D | $\underline{95.46} \pm 5.02$ |
| | Mamba | $93.55 \pm 4.53$ |
| | GRU | $93.57 \pm 4.22$ |
| | POYO | $94.86 \pm 3.53$ |
| | POSSM-S4D | $92.11 \pm 6.66$ |
| | POSSM-Mamba | $92.95 \pm 4.96$ |
| | POSSM-GRU | $\mathbf{95.82} \pm 3.41$ |
| PRETRAINED ON NHP | POYO-1 (FT) | $95.82 \pm 3.12$ |
| | o-POSSM-S4D (FT) | $97.25 \pm 2.88*$ |
| | o-POSSM-Mamba (FT) | $\mathbf{97.73} \pm 2.13**$ |
| | o-POSSM-GRU (FT) | $\underline{97.37} \pm 2.32*$ |

## 3.3 Human speech

Finally, we evaluated POSSM on the task of human speech decoding. Unlike the reaching and handwriting tasks, which involved fixed-length context windows, speech decoding involves modelling variable-length phoneme sequences that depend on both the length of the sentence and the individual's speaking pace. We used a public dataset [3, 33] consisting of 24 sessions in which a human participant with speech deficits attempted to speak sentences that appeared on a screen (Figure 3h). Neural activity was recorded from four 64-channel microelectrode arrays that covered the premotor cortex and Broca's area. Multi-unit spiking activity was extracted and binned at a resolution of 20 ms, where the length of each trial ranged from 2 to 18 seconds. This poses a problem for Transformers like POYO that were designed for 1 s contexts, as the quadratic complexity of the attention mechanism would result in a substantial increase in computation for longer sentences.

Table 3: **Human speech decoding results.** We report the average phoneme error rate (PER) over 3 seeds. All SDs $< 0.1\%$.

| Method | PER (%) ↓ |
|---|---|
| GRU (no aug.) | 39.16 |
| POSSM-GRU (no aug.) | **29.70** |
| GRU | 30.06 |
| S4D | 35.99 |
| Mamba | 32.19 |
| POSSM-GRU | **27.32** |
| GRU (mult.) | 21.74 |
| POSSM-GRU (mult.) | **19.80** |

Although both uni- and bi-directional GRUs were used in the original study [3, 33], we focused primarily on the causal, uni-directional case, as it is more relevant for real-time decoding. In line with Willett et al. [3], we z-scored the neural data and added Gaussian noise as an augmentation. We used the phoneme error rate (PER) as our primary evaluation metric. While prior work has successfully incorporated language models to leverage textual priors, we leave this as a future research direction, instead focusing here on POSSM's capabilities.

Previous work [33] has shown that Transformer-based decoders perform poorly on this task compared to GRU baselines. Here, we demonstrate the value of instead integrating attention with a recurrent model by using POSSM, specifically with a GRU backbone. However, since only normalized spike counts (and not spike times) were available in the dataset, we were unable to use the POYO-style tokenization as-is. Instead, we treated each multi-unit channel as a neural unit and encoded the normalized spike counts with value embeddings. Furthermore, we replaced the output cross-attention module with a 1D strided convolution layer to control the length of the output sequence. This

approach significantly reduced the number of model parameters compared to the GRU baseline, which used strided sliding windows of neural activity as inputs instead.

We found that a two-phase training procedure yielded the best results. In the first phase, we trained the input cross-attention module along with the latent and unit embeddings by reconstructing the spike counts at each individual time bin. In the second phase, we trained the entire POSSM model on the target phoneme sequences using Connectionist Temporal Classification (CTC) [34] loss.

**Results.** POSSM achieved a significant improvement over all other baselines, as shown in Table 3. Notably, POSSM maintained comparable performance even without the Gaussian noise augmentation, while the performance of the baseline GRU was greatly impaired under the same conditions. Furthermore, we show in preliminary experiments with multiple input modalities (i.e., both spike counts and spiking-band powers) that POSSM yet again outperforms the baseline. These results illustrate the robustness of the POSSM architecture to variability in data preprocessing and its flexibility with respect to input modalities, further strengthening its case as a feasible real-world decoder.

## 4 Related Work

**Neural decoding.** Neural decoding for continuous tasks such as motor control in BCIs was traditionally accomplished using statistical models such as the Kalman filter [35–38]. While these models are reliable and perform well for specific users and sessions, they require careful adaptation to generalize to new days, users, and even tasks. Such adaptation is typically accomplished using model fitting approaches to estimate the Kalman filter parameters [38–40], a process that requires considerable new training data for each application. Traditional approaches to multi-session neural decoding often consist of learning the decoder's parameters on a specific day or session (e.g., the first day), followed by learning models to align the neural activity on subsequent days to facilitate generalization [41–43]. Given the recent availability of large, public neural recordings datasets [3, 23, 28–30], modern neural decoding approaches have attempted to leverage advances in large-scale deep learning to build data-driven BCI decoders [11–13, 44–49]. For example, in the context of decoders for neuronal spiking activity, NDT [44] jointly embeds spikes from a neural population into a single token per time bin, spatiotemporal NDT (STNDT) [50] separately tokenizes across units and time and learns a joint representation across these two contexts, NDT-2 [11] tokenizes spatiotemporal patches of neural data akin to a ViT [51], and POYO [12] eschews bin-based tokenization, opting to tokenize individual spikes and using a PerceiverIO [22] Transformer backbone to query behaviours from within specific context windows. While several of these works excel at neural decoding, they do not focus on enabling generalizable, online decoding in spike-based BCIs and closed-loop protocols. The BRAND platform [8] enables the deployment of specialized deep learning models in real-time closed-loop brain-computer interface experiments with invasive recordings, demonstrating suitable low-latency neural decoding in other models such as LFADS [52]. Finally, we note other ML methods for neural data processing other than direct behaviour decoding. Contrastive learning methods that aim to identify joint latent variables between neural activity and behaviour can be useful for decoding but work is needed for online use [53]. Diffusion-based approaches are promising for jointly forecasting neural activity and behaviour, but again are not readily suited for online use [54].

**Hybrid attention-recurrence models.** Several works have attempted to combine self- and cross-attention layers with recurrent architectures, usually with the goal of combining the expressivity of attention over shorter timescales with the long-term context modelling abilities of recurrent models such as structured SSMs [15–17]. While traditional SSMs have been used for several neuroscience applications [55–58], modern SSMs and hybrid models remain underexplored in the field. Didolkar et al. [59] propose an architecture comprising Transformer blocks which process information at faster, shorter timescales while a recurrent backbone integrates information from these blocks over longer timescales for long-term contextual modeling. A similar approach is the Block-Recurrent Transformer [60], wherein a recurrent cell operates on a block of tokens in parallel, thus propagating a block of state vectors through timesteps. Pilault et al. [61] propose the Block-State Transformer architecture, which introduces a layer consisting of an SSM to process long input sequences, the outputs of which are sent in blocks to several Transformers that process them in parallel to produce output tokens. Furthermore, several recent works on high-throughput language modelling [19–21] have leveraged hybrid models, where self-attention layers are replaced with SSM blocks to take advantage of their subquadratic computational complexity.

# 5 Discussion

**Summary.** We introduced POSSM, a scalable and generalizable hybrid architecture that pairs spike tokenization and input-output cross-attention with SSMs. This architecture enables efficient online decoding applications, achieving state-of-the-art performance for several neural decoding tasks while having fewer parameters and faster inference compared to fully Transformer-based approaches. Our model achieves high performance with millisecond-scale inference times on standard workstations, even without GPUs, making it suitable for real-time deployment in clinical settings.

A key contribution of this work is demonstrating, to our knowledge, the first successful cross-species transfer of learned neural dynamics for a deep learning-based decoder – from NHP motor cortex to human clinical data (see [11, 62] for related efforts). This outlines a solution to a major clinical hurdle, where obtaining sufficient data for large-scale modelling is challenging or impossible in patient populations. We demonstrate that we can leverage the large corpus of existing non-human experimental data to improve personalized clinical outcomes by finetuning pretrained models.

**Future directions and applications.** The POSSM architecture is applied here for motor neural decoding, but it can be adapted to achieve a variety of outcomes. Our current model focuses on processing spiking data from implanted arrays in the motor cortex of monkeys and human clinical trial participants. However, our hybrid architecture is flexible and could readily accommodate data of other neural data modalities through a variety of proven tokenization schemes (e.g., Calcium imaging [13], EEG [45, 47, 63]). While in the present work we focus on decoding of behavioural timestamps immediately following our input time chunks, our hybrid architecture is well-suited towards forecasting over longer timescales. Further, in the future we plan to explore the ability of POSSM to learn and generalize across different regions beyond the motor cortex.

Ultimately, we envision POSSM as a first step towards a fast, generalizable neural foundation model for various neural interfacing tasks, with downstream applications such as clinical diagnostics and the development of smart, closed-loop neuromodulation techniques that link predicted states to optimized neural stimulators (e.g., [4, 5]). Future steps include multimodal pretraining and decoding [22, 49, 64, 65] as well as a principled self-supervised pretraining scheme. By enabling efficient inference and flexible generalization through transfer learning, POSSM marks a new direction for general-purpose neural decoders with real-world practicality.

## Broader Impact

This research could potentially contribute to the development of neural decoders that are not only accurate but also amenable to deployment in online systems, such as those found in neuroprosthetics and other brain-computer interfaces. In addition to POSSM's general performance, it reduces the need for extensive individual calibration due to the pretraining/finetuning scheme. Additionally, the cross-species transfer results on the handwriting task suggest that patients with limited availability of data could benefit from models pretrained with larger datasets from different species. While the potential downstream applications of POSSM are exciting, it is important to consider the ethical concerns that exist for any medical technology, including but not limited to data privacy and humane data collection from animals. Strict testing should be implemented before deployment in any human-related setting.

## Acknowledgments and Disclosure of Funding

The authors would like to thank Patrick Mineault, Blake Richards, Ayesha Vermani, and Alexandre André for support and feedback. MA acknowledges support from the NSF AI Institute for Artificial and Natural Intelligence. MGP acknowledges support of a Future Leaders award from the Brain Canada Foundation and a J1 Chercheurs-boursiers en intelligence artificielle from the Fonds de recherche du Québec – Santé. GL acknowledges support from NSERC Discovery Grant RGPIN-2018-04821, the Canada Research Chair in Neural Computations and Interfacing, a Canada-CIFAR AI Chair, IVADO, and the Canada First Research Excellence Fund. The authors also acknowledge the support of computational resources provided by Mila, the Digital Research Alliance of Canada, and NVIDIA that enabled this research.

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

# Supplementary Material

## A  Dataset Details

Table 4: **Summary of neural datasets used for pretraining and evaluation.** Values represent totals across training, validation, and testing subsets of the pretraining and evaluation datasets.

| | Dataset | Regions | Tasks | # Indiv. | # Sess. | # Units | # Spikes | # Bhvr. |
|---|---|---|---|---|---|---|---|---|
| TRAINING | Perich et al. [28] | M1, PMd | CO, RT | 2 | 93 | 9040 | 96.8M | 12.7M |
| | O'Doherty et al. [29] | M1, S1 | RT | 2 | 44 | 14 899 | 87.8M | 10.4M |
| | Churchland et al. [30] | M1, PMd | CO | 2 | 10 | 1911 | 739M | 85.0M |
| | NLB Maze [31] | M1, PMd | Maze | 1 | 1 | 182 | 3.64M | 6.81M |
| EVALUATION | Perich et al. [28] Monkey C | M1, PMd | CO | 1 | 2 | 520 | 4.65M | 221K |
| | Perich et al. [28] Monkey T | M1, PMd | CO | 1 | 6 | 336 | 1.53M | 589K |
| | Perich et al. [28] Monkey T | M1, PMd | RT | 1 | 6 | 349 | 1.59M | 551K |
| | Flint et al. [66] Monkey C | M1 | CO | 1 | 5 | 957 | 7.88M | 318K |
| | NLB Area2 [31] | S1 | CO | 1 | 1 | 65 | 222K | 50.7K |

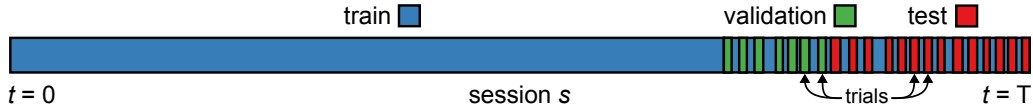

Figure 5: **Session-wise data splits for training, validation, and testing.** Per session (across all datasets), 10% of the trials were used for validation and 20% were used for testing. The remaining data, including inter-trial segments, was used for training.

**Dataset details.** For all our experiments, we attempted to minimize dataset preprocessing, in line with POYO [12]. The various datasets used during pretraining were collected from different labs using different experimental setups, effectors, etc. However, we did not standardize the datasets in any way and only excluded behavioural data based on a simple thresholding of acceleration values to remove outliers. Thus, like POYO: (1) we did not filter units or reject multi-units; (2) we did not apply identical spike sorting algorithms for all datasets (our training data consists of a mix of threshold crossing units and spike sorted single units); and (3) we did not resample or process the velocity, i.e., the behavioural output to be decoded. Details on the datasets used and session-wise data splits are provided in Table 4 and Figure 5 respectively. Note that for the NLB Area2 dataset [31], we did not include perturbed trials.

## B  Architectural Details

### B.1  Tokenization

For all our models, we used POYO-style tokenization [12], albeit with one key difference: unlike POYO, we did not provide delimiter tokens (i.e., [START] and [END] tokens) to the model. This reduced the total number of tokens passed as input to the model, thus increasing efficiency, but did not negatively impact decoding performance in our experiments.

### B.2  Recurrent backbones

In this section, we provide additional details on each of the recurrent backbones used by POSSM. It is important to note that POSSM is theoretically agnostic to the type of recurrent architecture.

**Gated recurrent unit (GRU).** Like traditional recurrent neural networks, the GRU [14] maintains a hidden state $\mathbf{h}_t \in \mathbb{R}^d$ that is governed by the input at the current timestep $\mathbf{x}_t \in \mathbb{R}^n$ and the previous hidden state $\mathbf{h}_{t-1}$. At each timestep, the network calculates an update gate $\mathbf{z}_t$ and a reset gate $\mathbf{r}_t$,

defined as follows:

$$\mathbf{z}_t = \sigma(\mathbf{W}_z\mathbf{x}_t + \mathbf{U}_z\mathbf{h}_{t-1} + \mathbf{b}_z), \ \ \mathbf{r}_t = \sigma(\mathbf{W}_r\mathbf{x}_t + \mathbf{U}_r\mathbf{h}_{t-1} + \mathbf{b}_r),$$

where $\mathbf{W}_z, \mathbf{W}_r \in \mathbb{R}^{d\times n}$, $\mathbf{U}_z, \mathbf{U}_r \in \mathbb{R}^{d\times d}$, and $\mathbf{b}_z, \mathbf{b}_r \in \mathbb{R}^d$ are the input weights, hidden state weights, and biases, respectively. $\sigma$ is the sigmoid activation function. A candidate hidden state is then generated:

$$\tilde{\mathbf{h}}_t = \tanh(\mathbf{W}_h\mathbf{x}_t + \mathbf{U}_h(\mathbf{r}_t \odot \mathbf{h}_{t-1}) + \mathbf{b}_h),$$

where $\mathbf{W}_h \in \mathbb{R}^{d\times n}, \mathbf{U}_h \in \mathbb{R}^{d\times d}, \mathbf{b}_h \in \mathbb{R}^d$ are learnable parameters and $\odot$ is the Hadamard product.

The final hidden state is calculated by weighing the candidate and previous hidden states by the update and reset gates:

$$\mathbf{h}_t = (1 - \mathbf{z}_t) \odot \mathbf{h}_{t-1} + \mathbf{z}_t \odot \tilde{\mathbf{h}}_t.$$

**Diagonal structured state-space models (S4D).** Structured state-space models (S4) [15] are built on the theory of linear time-invariant SSMs. Specifically, the hidden state $\mathbf{x}(t) \in \mathbb{R}^d$ and output $y(t) \in \mathbb{R}$ are calculated as follows:

$$\dot{\mathbf{x}}(t) = \mathbf{A}\mathbf{x}(t) + \mathbf{B}u(t), \ \ y(t) = \mathbf{C}\mathbf{x}(t),$$

where $\mathbf{A} \in \mathbb{R}^{d\times d}, \mathbf{B} \in \mathbb{R}^{d\times 1}$, and $\mathbf{C} \in \mathbb{R}^{1\times d}$ are weight matrices and $u(t) \in \mathbb{R}$ is the input at time $t$. An additional parameter, $\Delta$, denotes the step size or the time-resolution of the input. For inputs with multiple channels, the SSM is applied independently to each channel. To be applied to a discrete input sequence $(u_0, u_1, \ldots)$ rather than a continuous function $u(t)$, the above continuous-time SSM can be discretized through a transformation of its parameters $(\mathbf{A}, \mathbf{B}, \mathbf{C}, \Delta)$, yielding $(\bar{\mathbf{A}}, \bar{\mathbf{B}}, \mathbf{C})$ [15]. Unlike traditional RNNs that can only process sequences one element at a time, S4's linear nature allows it to formulate its output as a discrete convolution post discretization:

$$y_t = \sum_{k=0}^{L} \mathbf{K}_k u_{t-k},$$

where $L$ is the length of the sequence and $\mathbf{K}$ is a convolutional kernel parameterized as such:

$$\mathbf{K}_k = \mathbf{C}\bar{\mathbf{A}}^k\mathbf{B}.$$

This powerful property allows the computation to be parallelized across elements of the sequence during training, resulting in faster training than non-linear RNNs of the same size. Finally, in this work, we use the Diagonal S4 version (S4D), which replaces the traditional HiPPO initialization [67] with a diagonal-only version which is faster yet has been shown to retain similar performance.

**Mamba.** Building on the SSM neural network introduced by S4, Mamba [16] also aims to be a recurrent architecture that allows for parallelizable inference during training. However, unlike S4, the model learns dynamic weight matrices $\mathbf{B}_t$ and $\mathbf{C}_t$ that are now parameterized by position $t$ in the sequence (and are functions of the input). This allows Mamba to selectively compress information into the hidden state, essentially discarding information that is deemed unimportant.

### B.3 Non-human primate reaching

The hyperparameters changed between the single-session model and the pretrained (o-POSSM) models are the input dimensionality (i.e., size of the spike and latent tokens), RNN hidden dimensionality, and the number of RNN layers. While a set of hyperparameters was kept constant across single-session and multi-session models (one set for each), the number of parameters that the model has varies based on the choice of recurrent backbone. We find that S4D is the smallest and Mamba is the largest when these three hyperparameters are kept constant. Details are provided in Table 5.

### B.4 Human handwriting

**Baselines.** The statistical baseline PCA-KNN was implemented following Willett et al. [23]. A Gaussian kernel was initially used to smooth the spikes, followed by PCA to reduce the dimensionality to 15. Then, 7 time-warp factors were used to resample the PC-projected neural activity, and Euclidean distances were calculated between the results of each time-warp factor and all other points in the dataset. Finally, the minimum time-warp distances over all 7 factors were used to classify characters using a k-nearest neighbours (KNN) classifier. The only difference here is that we considered 1.6 s intervals of neural activity. For more details on this method, please refer to the original publication.

Table 5: **Hyperparameters and model sizes of POSSM variants used for NHP reaching tasks.**

| Backbone | SS / FT | Input Dim. | RNN Hidden Dim. | # RNN Layers | # Params. |
|----------|---------|------------|-----------------|--------------|-----------|
| S4D | SS | 64 | 256 | 1 | 0.41M |
| | FT | 256 | 512 | 4 | 4.56M |
| GRU | SS | 64 | 256 | 1 | 0.47M |
| | FT | 256 | 512 | 4 | 7.96M |
| Mamba | SS | 64 | 256 | 1 | 0.68M |
| | FT | 256 | 512 | 4 | 8.96M |

**POSSM.** We used the same POSSM architecture as the NHP reaching tasks.

### B.5 Human speech

**Baselines.** We adopted the same hyperparameters as mentioned in Willett et al. [3] for the baseline GRU, including block-wise normalization and two sources of Gaussian noise injection. Sliding windows of 32 bins of normalized spike counts with a stride of 4 bins were used at the input for both uni-directional and bi-directional models. For uni-directional and bi-directional GRUs, the number of parameters were 55M and 133M, respectively. We also introduce two more binning-based baselines by replacing the GRU backbone of the model with Mamba and S4D.

**POSSM.** We used POSSM with a GRU as the recurrent backbone (uni-directional and bi-directional according to the problem setup) for this task. The encoder was a cross-attention module with a single head, followed by a self-attention module with 2 heads, operating strictly within each 20 ms bin. We used an input dimension of 64 and 4 latents per bin. At the output of the encoder, we concatenated the latents from the same bin, resulting in a 256-dimensional vector that served as the input to the GRU. As mentioned in the previous section, the baseline method processed sliding windows of neural activity inputs, consisting of $k$ bins (usually 14 or 32) with stride $s$ (usually 4). Values of $k$ and $s$ were chosen to control the length of the output sequence, with larger values of $k$ being shown to help performance [3]. In this paper, we instead set both $k$ and $s$ as 1 and utilized a convolutional layer at the output to control the length of output sequences. This ensured that the emission frequency of output phonemes from POSSM was identical to that of the baselines. The baseline method also introduced a linear layer to map the concatenation of all $k$ bins to a fixed hidden dimension, which quickly became computationally expensive – especially when $k$ was large. For POSSM, avoiding the use of sliding windows led to a sizable parameter reduction (e.g., 24M fewer parameters than a GRU with the above setup). For both uni-directional and bi-directional POSSM, the parameter counts were 32M and 86M, respectively.

## C Training Details

### C.1 Non-human primate reaching

As the prediction target is a two-dimensional time series of either hand or cursor velocity, the loss function was chosen to be mean squared error. We increased the weight of the loss for centre-out reaching segments by a factor of 5, following POYO. A maximum of 100 behaviour values were provided as targets for each 1 s training sample. If the dataset had a behavioural sampling rate greater than 100 Hz, 100 behaviour samples were randomly chosen within the second.

**Baselines.** We considered four single-session binning-based baselines for the reaching tasks: MLP, GRU, S4, and Mamba. We used 50 ms bins to extract spike counts for these models, with step sizes of 50 ms and 10 ms for the MLP and the other models, respectively. All baselines were trained using the AdamW optimizer [68] with a batch size of 128, a base learning rate of 0.004, and a cosine scheduler for a total of 500 epochs.

In addition to the aforementioned models, both single-session and pretrained versions of POYO were used as baselines. Following Azabou et al. [12], we used a context window of 1 s for POYO (and a step size of 50 ms during evaluation, as described in Section 3.1). The model was trained using the LAMB

optimizer [69] with a batch size of 128, base learning rates of 0.004 and 0.002 for single-session and pretrained models respectively, and a cosine scheduler for a total of 500 epochs. During training, we also applied a data augmentation scheme called unit dropout [12, 70], where we randomly drop out a subset of units from the input along with all their spikes for each iteration. Both unit identification and full finetuning used the same training hyperparameters as the single-session models, except with a constant learning rate. For full finetuning, unit identification was first performed for 100 epochs before the rest of the model was unfrozen and trained for another 400 epochs. All models were trained for a total of 500 epochs.

Finally, for NDT-2, we used the code provided by Ye et al. [11] and their default hyperparameters. However, we pretrained and evaluated NDT-2 on the same datasets and splits used for POSSM.

**POSSM.** Single-session models were trained on a single NVIDIA RTX8000 GPU using LAMB with a batch size of 128, a base learning rate of 0.004, and a cosine scheduler. Multi-dataset pretraining was done on four NVIDIA H100 GPUs using LAMB with a batch size of 256, a base learning rate of 0.002, and a cosine scheduler. The unit dropout data augmentation scheme mentioned previously was employed during training. Both unit identification and full finetuning used the same training hyperparameters as the single-session models, except with a constant learning rate. For full finetuning, unit identification was first performed for 100 epochs before the rest of the model was unfrozen and trained for another 400 epochs. All models were trained for a total of 500 epochs. Single-session models took less than 30 minutes to train, while multi-dataset pretraining took around 36 hours.

### C.2 Human handwriting

**Baselines.** For the PCA-KNN baseline, we performed hyperparameter tuning on a validation set for each session, selecting the number of neighbours from a range of 2 to 14. For the GRU, S4, and Mamba, we provided non-overlapping 50 ms bins of spike counts as inputs.

**POYO and POSSM.** In this task, we used the unit dropout data augmentation scheme for POYO and POSSM. As for POYO, aside from unit dropout, we applied an additional augmentation scheme called random time-scaling, where the timestamps are randomly scaled at each iteration.

A gradual unfreezing strategy was used during finetuning for the handwriting task, where only the unit embeddings, session embeddings, and the linear readout layer for the task were trained at the beginning. Note that unfreezing the linear readout layer differentiates the finetuning strategy for the handwriting task from the one used for the reaching tasks described previously. Training the linear layer was necessary as handwriting decoding was not present in the pretraining datasets. We finetuned POYO-1 for 600 epochs (unfrozen at epoch 40), while o-POSSM was finetuned for a total of 1000 epochs (unfrozen at epoch 50). All training hyperparameters for POYO followed the suggestions of its original authors.

For POYO and POSSM trained from scratch, as well as finetuned POYO-1, training was performed on a single NVIDIA RTX8000 GPU using LAMB with a batch size of 256, a base learning rate of 0.00256, and a cosine scheduler. When finetuning o-POSSM, the base learning rate was 0.008 instead. We trained the models with three random seeds and reported the mean accuracy.

### C.3 Human speech

**Baselines.** All baselines for speech decoding were trained on a single NVIDIA RTX8000 GPU using AdamW with a batch size of 64 and a base learning rate of 0.02. For uni-directional models, a linear decaying scheduler was applied to the base learning rate, while for bi-directional models, the learning rate was kept constant.

**POSSM.** Both the uni-directional and bi-directional versions of POSSM were trained on a single NVIDIA A100 GPU (80GB) with a batch size of 16. We conducted a hyperparameter sweep on the learning rate schedule via cross-validation on the training split. In the uni-directional case, we adopted a base learning rate of 0.0001 with a cosine scheduler. In the bi-directional case, we used a base learning rate of 0.00004 with a two-phase one-cycle scheduler where the learning rate was first increased to 0.00008 at epoch 50 then reduced with cosine annealing.

We trained all models for 100 epochs and with three random seeds, and reported the mean accuracy.

# D Additional Results

## D.1 Performance on NHP reaching with 20 ms time chunks

We ran additional experiments comparing single-session models working with contiguous time chunks of 20 ms as opposed to 50 ms. As shown in Table 6, we do not see drops in POSSM's performance with these smaller time chunks, and it continues to perform better than or on par with baseline methods. In practice, given POSSM's low inference times, it is possible to run POSSM in real-time with 50 ms time chunks and a step size of 20 ms, if a quicker prediction frequency is desired.

Table 6: **Behavioural decoding results on NHP reaching tasks with 20 ms time chunks.** Best performing models are in boldface (1st) and underlined (2nd).

|  | Method | Same animal, other days | | New animal | | New dataset |
|---|---|---|---|---|---|---|
|  |  | C – CO 2016 (2) | C – CO 2010 (5) | T – CO (6) | T – RT (6) | H – CO (1) |
| FROM SCRATCH | MLP | 0.9269 ± 0.0044 | 0.5842 ± 0.2052 | 0.7940 ± 0.0341 | 0.6082 ± 0.3014 | 0.7602 |
|  | S4D | 0.9313 ± 0.0038 | 0.6842 ± 0.1017 | 0.8597 ± 0.0275 | 0.7120 ± 0.0764 | 0.4554 |
|  | Mamba | 0.9283 ± 0.0096 | 0.6840 ± 0.0936 | 0.7318 ± 0.0426 | 0.6653 ± 0.0978 | 0.7045 |
|  | GRU | 0.9435 ± 0.0069 | 0.7742 ± 0.0964 | 0.8389 ± 0.0248 | 0.7414 ± 0.0426 | **0.8101** |
|  | POSSM-S4D-SS | 0.9470 ± 0.0016 | 0.7611 ± 0.1179 | **0.8777** ± 0.0228 | 0.7369 ± 0.0691 | 0.7958 |
|  | POSSM-Mamba-SS | 0.9490 ± 0.0059 | 0.7691 ± 0.0786 | 0.8613 ± 0.0121 | 0.7300 ± 0.0719 | 0.7065 |
|  | POSSM-GRU-SS | **0.9514** ± 0.0010 | **0.7780** ± 0.0980 | 0.8724 ± 0.0190 | **0.7429** ± 0.0708 | 0.7816 |
| PRETRAINED | o-POSSM-S4D (UI) | 0.9589 ± 0.0025 | 0.7835 ± 0.0839 | 0.8661 ± 0.0201 | 0.7256 ± 0.0740 | 0.8402 |
|  | o-POSSM-Mamba (UI) | 0.9588 ± 0.0006 | 0.7175 ± 0.1373 | 0.8483 ± 0.0247 | 0.7226 ± 0.0870 | 0.8102 |
|  | o-POSSM-GRU (UI) | 0.9551 ± 0.0070 | 0.7837 ± 0.0916 | 0.8619 ± 0.0304 | 0.7323 ± 0.0812 | 0.8087 |
|  | o-POSSM-S4D (FT) | 0.9601 ± 0.0062 | 0.8052 ± 0.0887 | **0.8991** ± 0.0229 | 0.7543 ± 0.0672 | **0.8834** |
|  | o-POSSM-Mamba (FT) | **0.9605** ± 0.0016 | 0.8034 ± 0.0971 | 0.8946 ± 0.0168 | 0.7508 ± 0.0816 | 0.8639 |
|  | o-POSSM-GRU (FT) | 0.9580 ± 0.0040 | **0.8110** ± 0.0753 | 0.8908 ± 0.0214 | **0.7561** ± 0.0727 | 0.8588 |

## D.2 Effects of context length on handwriting decoding

In Figure 6, we plot imagined handwriting decoding accuracy against the context length that POSSM receives. This shows that POSSM can make accurate predictions in a causal manner, given sufficient context. More importantly, these results showcase POSSM's applicability to real-time applications, with time series data fed into the neural decoder in a streamlined fashion.

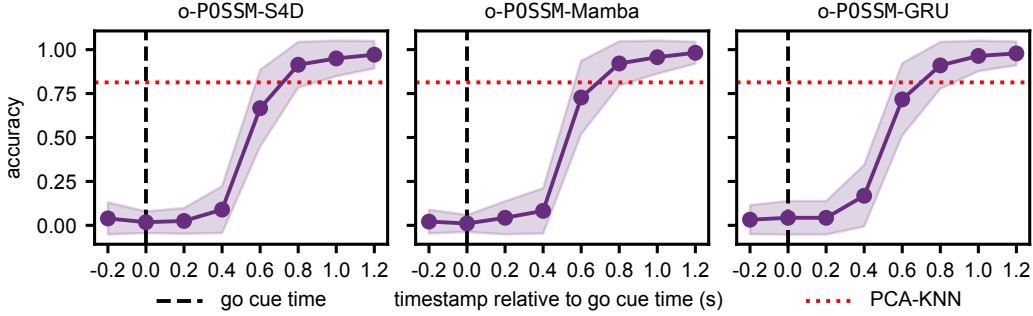

Figure 6: **Handwriting decoding accuracy given various context lengths.** We plot mean accuracies and 99% confidence intervals for each finetuned o-POSSM model, given an increasing amount of neural activity as context. For each context length, POSSM makes a prediction at the last timestamp. The plot's x-axis is aligned with the "go" cue time.

## D.3 Speech decoding with bi-directional models

We ran the bi-directional versions of the top competitors identified in the uni-directional case. As shown in Table 7, both with and without noise augmentation, bi-directional models demonstrated a consistent improvement of 2–3% over their uni-directional counterparts. This is expected as

bi-directional models can utilize the full temporal sequence of neural activity for each phoneme prediction. Similarly to the uni-directional case, we observed that POSSM outperformed the baseline even without noise augmentation.

Table 7: **Human speech decoding results with bi-directional model backbones.** We report the phoneme error rate (PER) averaged over 3 seeds. All SDs $< 0.1\%$.

| Method | PER (%) $\downarrow$ |
|---|---|
| BiGRU (no aug.) | 37.18 |
| POSSM-BiGRU (no aug.) | **26.62** |
| BiGRU | 27.86 |
| POSSM-BiGRU | **25.80** |

### D.4 Decoding behaviour from non-motor neural activity

To validate the applicability of POSSM to decoding behaviour from other, non-motor brain regions, we ran a preliminary experiment with POSSM on 5 randomly selected sessions from the Allen Visual Behaviour Neuropixels dataset [71], which tracked spiking activity in mice. Here, we attempted to decode running speed from multi-region neural activity (visual cortex and several subcortical areas, including visual thalamic areas, hippocampus, and midbrain). We found that a single-session POSSM-GRU achieved an $R^2$ of $0.9295 \pm 0.0326$, averaged over 5 sessions.

### D.5 Predicting future behaviour at each time chunk

In order to demonstrate the flexibility of our output cross-attention module, we trained single-session POSSM models to predict cursor velocities at timestamps that were 50 ms or 100 ms beyond the end of each time chunk (denoted as "lags"). Thus, we queried the output module with timestamps beyond those associated with the input time chunk. Our results in this setting were on par with, or only slightly lower than, single-session POSSM performance with no lags (Table 8).

Table 8: **Results on decoding cursor velocities at timestamps beyond the end of each time chunk.** We consider lags of 50 ms and 100 ms, and report the mean $R^2 \pm$ SD over sessions.

| Method | C – CO 2016 (2) | T – RT (6) |
|---|---|---|
| POSSM-GRU-SS (50 ms lag) | $0.9495 \pm 0.0045$ | $0.7498 \pm 0.0874$ |
| POSSM-GRU-SS (100 ms lag) | $0.9368 \pm 0.0027$ | $0.7252 \pm 0.0813$ |

### D.6 Ablating spike timing information

A key advantage of POYO-style tokenization, apart from facilitating pretraining and generalization, is that it retains spike timing information unlike binning-based methods. To specifically test whether this scheme yielded performance advantages over binning, we trained single-session POSSM models without precise spike timing information, instead providing bin-level timestamps for all spikes. We found that models receiving spike timing information consistently outperformed those receiving only bin-level timestamps (Table 9), showing the advantage of precise spike times for neural decoding.

Table 9: **Behavioural decoding results with and without precise spike times.** We report the mean $R^2 \pm$ SD over sessions.

| Method | C – CO 2010 (5) | T – RT (6) | H – CO (1) |
|---|---|---|---|
| POSSM-Mamba-SS (w/ spike times) | $0.7959 \pm 0.0772$ | $0.7418 \pm 0.0790$ | $0.8350$ |
| POSSM-Mamba-SS (w/ bin-level times) | $0.7040 \pm 0.0903$ | $0.7275 \pm 0.0734$ | $0.5568$ |

### D.7 Using feature mixer MLPs

While we did not use feature mixer MLPs after cross-attention and SSM layers in our experiments, we conducted an experiment to test whether using them improved decoding performance. As shown in Table 10, performance with feature mixers was comparable to or slightly better than our existing single-session models. However, these improvements come at the cost of increased model size and inference times. We leave a more detailed investigation on this for future work.

Table 10: **Behavioural decoding results with and without feature mixer MLPs.** We report the mean $R^2 \pm$ SD over sessions.

| Method | C – CO 2016 (2) | T – RT (6) |
|---|---|---|
| POSSM-Mamba-SS (w/ mixers) | $0.9560 \pm 0.0018$ | $0.7487 \pm 0.0651$ |
| POSSM-Mamba-SS (no mixers) | $0.9550 \pm 0.0003$ | $0.7418 \pm 0.0790$ |

### D.8 Using a recurrent encoder instead of cross-attention

We conducted an experiment with single-session POSSM models using a recurrent encoder (GRU) to encode spike tokens instead of the input cross-attention module. We maintained the same input configuration as with the cross-attention, providing contiguous, non-overlapping 50 ms time chunks as inputs to the model. The encoder's hidden state was reinitialized for each chunk, although in practice a state could be maintained across chunks for streaming neural activity. We did not use any positional embeddings for the spikes, but they were ordered by spike times. Our results showed that models with a recurrent encoder underperformed the standard POSSM models, but were superior to those not receiving precise spike-timing information (Table 11).

Table 11: **Behavioural decoding results with a recurrent encoder vs. a cross-attention encoder with and without precise spike timing information.** We report the mean $R^2 \pm$ SD over sessions.

| Method | C – CO 2016 (2) | T – RT (6) |
|---|---|---|
| POSSM-GRU-SS (w/ GRU encoder) | $0.9396 \pm 0.0066$ | $0.7334 \pm 0.0611$ |
| POSSM-GRU-SS (w/ spike times) | $0.9549 \pm 0.0012$ | $0.7687 \pm 0.0669$ |
| POSSM-GRU-SS (w/o spike times) | $0.9026 \pm 0.0135$ | $0.7207 \pm 0.0632$ |

### D.9 LoRA finetuning as an effective alternative to unit identification

In a preliminary experiment with o-POSSM-GRU, we showed that finetuning embeddings on held-out sessions with low-rank adapters (LoRA) [72] could be an effective alternative to unit identification (Table 12). This finetuning scheme trained fewer parameters and was quicker compared to unit identification. A more detailed investigation is left for future work.

Table 12: **Behavioural decoding results with LoRA finetuning compared to UI.** We report the mean $R^2 \pm$ SD over sessions.

| Method | T – RT (6) | Avg. # params. finetuned |
|---|---|---|
| o-POSSM-GRU (LoRA, r = 16, $\alpha$ = 16) | $0.7478 \pm 0.0634$ | 9.16K |
| o-POSSM-GRU (UI) | $0.7464 \pm 0.0692$ | 15.6K |

