# OpenReview forum: "Generalizable, real-time neural decoding with hybrid state-space models"
_NeurIPS.cc/2025/Conference — NeurIPS 2025 poster_

### Official Review · Reviewer_Xwt5 · 2025-06-27

**Clarity:** 3
**Significance:** 4
**Originality:** 3
**Rating:** 5
**Confidence:** 4

**Summary:**

The paper introduces POSSM, a hybrid model architecture for spike-based neural decoding that builds upon the POYO-1 framework. While maintaining state-of-the-art (SOTA) performance, POSSM is designed to operate causally and with low latency, making it well-suited for real-time brain-computer interface (BCI) applications. Evaluations across diverse datasets highlight the model's robustness and generalization capabilities, with particularly notable results in cross-species knowledge transfer. This suggests that POSSM could represent an early step toward foundational models in neural decoding.

**Questions:**

Questions:
- In the first cross attention layer, why you don't use again a SSM so you don't need to process spikes by 50ms windows. You could also do that with a sliding window attention architecture. Furthermore, you could use a sliding window attention instead of an SSM, which could be an interesting extra comparison (I mention that because it seems that they perform very similarly as it was shown in MesaNet[von Oswald et al 2025]).
- In line 154, in the equation it would be nice to have the dimenssionality of the $W_k$ and $W_v$ matrices, I assume they are MxN.
- Something that was a bit unclear to me is whether you have feature mixers (a small MLP, or a multiplication with a gate like it happens in Mamba) after the attention or SSM layer. Feature mixers although a bit computationally expensive, they can improve the performance considerably.
- From Table 1, some models trained on single sessions perform better than pretrained models with UI (like POSSM-GRU) for the new dataset, do i read it, right? It's also interesting POSSM-SS models perform so much better than POYO-SS, do you have any idea why that is the case? It seems that the UI finetuning is not that useful for the SSM models. Maybe a possibility could be to do a LORA fine-tuning, which could be more efficient?

**Ethical Concerns:**

["NO or VERY MINOR ethics concerns only"]

**Final Justification:**

As I wrote from my initial review, I found that this paper is technically solid and is very interesting for the field. The authors clarified all my questions, and I am pleased to keep my initial score, as I explained in my official comment.

**Limitations:**

yes

**Paper Formatting Concerns:**

No issues with the paper format

**Quality:**

4

**Strengths And Weaknesses:**

Strengths:
- Achieves SOTA performance with low latency, which is crucial in neural decoding and BCI applications.
- They manange to extend the POYO-1 architecture, with the smart spike tokenization, allowing it to run causally and at the same time well below 10ms.

Weaknesses:
- The results in Table 2 are not well presented, because from the numbers except from PCA-KNN all the other result do not look to differ significantly. given that the plus-minus is the std (where if i am not wrong the $\pm$ is not reported).
- In section 4.3, if I understood correctly you first transform the spikes with some preprocessing and then in the first step you retrieve again spikes, which looks odd.

---

> ### Author Rebuttal · Authors · 2025-07-30
>
> We sincerely thank you for your positive assessment of our paper and insightful review! Below, we respond to your concerns and questions:
>
> > W1. The results in Table 2 are not well presented, because from the numbers except from PCA-KNN all the other result do not look to differ significantly. given that the plus-minus is the std (where if i am not wrong the $\pm$ is not reported).
>
> We apologize for the lack of clarity, we reported the arithmetic mean of mean performance across sessions $\pm$ the geometric mean of standard deviations across sessions, aggregated across seeds. We will update the caption to state this clearly. We also conducted a paired t-test between session-wise performances across seeds, between the best NHP-pretrained + finetuned o-POSSM and the best non-pretrained POSSM, where we found the results to be significant with p < 0.05 (p = 0.0152). Furthermore, comparing the mean performances across sessions for different seeds between all NHP-pretrained and non-pretrained models yielded p < 0.05 for 7/9 valid comparisons, p <= 0.1 for 2/9 cases (lower performing o-POSSM variants vs. non-pretrained POGRU). Thus, we show that the transfer result is statistically significant overall. We will update our paper to reflect this.
>
> > W2. In section 4.3, if I understood correctly you first transform the spikes with some preprocessing and then in the first step you retrieve again spikes, which looks odd.
>
> Apologies that this was not clear. We do not transform or preprocess the spikes; the data from the original source is already preprocessed through 20ms binning [1]. Therefore, we do not have access to raw spike timings. From preliminary experiments on other datasets, we found that having access to these raw spike timings can significantly improve performance (see table below). Additionally, given that the dataset is small, we found that our two-stage training curriculum yielded the best performance. Given access to more data and raw spikes, we believe this scheme would become unnecessary.
>
> |Dataset|POSSM-Mamba-SS w/ spike times|w/o spike times|
> |-|-|-|
> |C–CO 2010|0.7959 $\pm$ 0.0772|0.7040 $\pm$ 0.0903|
> |T–RT|0.7418 $\pm$ 0.0790|0.7275 $\pm$ 0.0734|
> |NLB Area2|0.8350|0.5568|
>
> > Q1. In the first cross attention layer, why you don't use again a SSM so you don't need to process spikes by 50ms windows. You could also do that with a sliding window attention architecture. Furthermore, you could use a sliding window attention instead of an SSM, which could be an interesting extra comparison (I mention that because it seems that they perform very similarly as it was shown in MesaNet).
>
> If we understand you correctly, using an SSM instead of the first cross-attention (CA) layer would mean that each token is processed by the model as each spike is recorded, i.e., one forward pass per token. This could perhaps allow for more fine-grained prediction timescales, but might be wasteful in terms of computational efficiency and is overall a harder problem because: (1) the sequences will become far longer and (2) the timing between spike tokens is highly irregular. Observing a window of activity for the whole population is important for the model to predict behaviour in dynamics tasks, which is why we use CA to process chunks of activity. Please let us know if we have misunderstood something or if you have an alternate implementation in mind, we would be happy to explore this.
>
> Sliding window attention (SWA) seems better suited to our tasks, and we have experimented with giving the model overlapping time chunks of a certain width and stride (e.g., 50ms and 20ms respectively) – performance is comparable to the contiguous non-overlapping window case.
>
> Replacing the recurrent backbone with SWA makes for an interesting additional comparison. We provide some results in the table below with one SWA layer in single session models. We would like to note that von Oswald et al. [2] show that smaller sliding windows do not perform very well. In our case, the performance of SWA does not match the best recurrent backbone – and we interestingly observe that larger window sizes perform slightly worse than smaller ones. This would have to be studied further, with hyperparameter tuning of the number of attention layers or heads, their dimensionality, dropout, etc. – which we will explore in the future.
>
> |Model|C–CO 2016|T–RT|
> |-|-|-|
> |Best POSSM-SS|0.9550 $\pm$ 0.0003|0.7687 $\pm$ 0.0669|
> |POSSM-SS-SWA(5)|0.9492 $\pm$ 0.0038|0.7188 $\pm$ 0.0966|
> |POSSM-SS-SWA(15)|0.9469 $\pm$ 0.0011|0.7043 $\pm$ 0.1117|
> |POSSM-SS-SWA(20)|0.9449 $\pm$ 0.0005|0.6736 $\pm$ 0.1282|
>
> > Q2. In line 154, in the equation it would be nice to have the dimenssionality of the $W_k$ and $W_v$ matrices, I assume they are MxN.
>
> Yes, thank you for the suggestion and apologies for not including this earlier – we have updated the paper to state the dimensionality explicitly.
>
> > Q3. Something that was a bit unclear to me is whether you have feature mixers (a small MLP, or a multiplication with a gate like it happens in Mamba) after the attention or SSM layer. Feature mixers although a bit computationally expensive, they can improve the performance considerably.
>
> We did not use mixer MLPs after the attention or SSM layers, mostly due to efficiency and strong performance even without them. While mixer MLPs provided marginal improvements in POSSM-SS models, they may be necessary at greater pretraining scale, so we will explore these in more detail in future work. A comparison with POSSM-Mamba-SS on the Perich et al. dataset [3] is given below, showing a slight increase in performance with mixer MLPs. We will add these results to our paper.
>
> |Type|C–CO 2016|T–CO|T–RT|
> |-|-|-|-|
> |No MLP|0.9550 $\pm$ 0.0003|0.8747 $\pm$ 0.0173|0.7418 $\pm$ 0.0790|
> |With MLP|0.9560 $\pm$ 0.0018|0.8828 $\pm$ 0.0164|0.7487 $\pm$ 0.0651|
>
> > Q4. From Table 1, some models trained on single sessions perform better than pretrained models with UI (like POSSM-GRU) for the new dataset, do i read it, right? It's also interesting POSSM-SS models perform so much better than POYO-SS, do you have any idea why that is the case? It seems that the UI finetuning is not that useful for the SSM models. Maybe a possibility could be to do a LORA fine-tuning, which could be more efficient?
>
> Thank you for this question, this was the case before. However, we now pretrained models on more data, including all 10 sessions from the Churchland et al. dataset [4] as opposed to just 2 earlier. With this, we significantly improved our UI results, and now they perform on par with or sometimes even better than POSSM-SS models (see table) on data from Perich et al. [3]. With these, we are able to finetune very few parameters overall for far less time and still achieve good performance.
>
> |Model|C–CO 2016|T–CO|T–RT|
> |-|-|-|-|
> |Best POSSM-SS|0.9550 $\pm$ 0.0003|0.8863 $\pm$ 0.0222|0.7687 $\pm$ 0.0669|
> |Best POSSM-UI|0.9607 $\pm$ 0.0036|0.8921 $\pm$ 0.0174|0.7464 $\pm$ 0.0692|
>
> LoRA is a great suggestion: we are exploring this in ongoing work involving SSL pretraining. For the sake of further compute efficiency and few-shot benefits on top of LoRA, it might be worth bootstrapping unit embeddings using existing pretrained ones (rather than starting from random for a new session or animal) – this direction requires further exploration. We believe this kind of LoRA-based UI pipeline is best suited to adaptation across days for the same animal, and especially when embedding multiunits (channels) rather than single neurons.
>
> We present some preliminary results below, where we assign existing trained unit embeddings for new units (in a new animal heldout from training), and finetune a LoRA for the embedding layer. We anticipate improvements with further hyperparameter tuning.
>
> |o-POSSM-GRU|T–RT|avg # params finetuned (range)|
> |-|-|-|
> |UI|0.7464 $\pm$ 0.0692|15.66K (10-19.5K)|
> |LoRA (r=16, $\alpha$=16)|0.7478 $\pm$ 0.0634|9.16K (8.8-9.4K)|
>
> It is unclear exactly why POYO-SS slightly underperforms POSSM-SS, although potential explanations include:
>
> * Prior work [5] finds that recurrent architectures are better-suited to neural decoding than transformers (caveat: smaller dataset scale).
> * Other work [6] shows that the inductive biases of SSMs allow them to learn delay embeddings better than transformers, thus enabling better performance on dynamics tasks (especially with less data and fewer parameters).
>
> We do find that POYO performs on par with or marginally better than o-POSSM on some multi-session experiments (e.g., Flint et al. dataset [7]). Deeper investigation into the latent representations learnt by these models is required to understand these differences in performance, and we are keen on exploring this in future work. For a preliminary study, please refer to our response to Reviewer eRCh, where we interpret the SSM hidden states.
>
> ---
>
> References:
>
> 1. Willett et al. "A high-performance speech neuroprosthesis." Nature (2023).
> 2. von Oswald et al. "MesaNet: Sequence Modeling by Locally Optimal Test-Time Training." arXiv:2506.05233 (2025).
> 3. Perich et al. "A Neural Population Mechanism for Rapid Learning." Neuron (2018).
> 4. Churchland et al. "Neural population dynamics during reaching." Nature (2012).
> 5. Willett et al. "Brain-to-Text Benchmark'24: Lessons Learned." arXiv:2412.17227 (2024).
> 6. Ostrow et al. "Delay embedding theory of neural sequence models." arXiv:2406.11993 (2024).
> 7. Flint et al. "Accurate decoding of reaching movements from field potentials in the absence of spikes." Journal of Neural Engineering (2012).
>
> ---
>
> Once again, thank you for your time, insightful suggestions, and kind assessment of our paper. We will endavour to include all these additional results in our paper and supplementary material. Overall, we hope this rebuttal addresses your concerns, and would be grateful if you would consider supporting our paper more strongly. We would be happy to answer any other questions during the discussion phase.

---

> > ### Comment · Reviewer_Xwt5 · 2025-08-04
> >
> > Thank you for your very detailed response and for clarifying/answering all of my raised points.
> > I have a small follow-up regarding Q1: I agree with you that running your model per spike would be quite wasteful, but you could (similarly to SWA) use a buffer and run the model every 20-50ms, but similarly to the SWA you don't break the continuity of input spikes. I am not sure what do you mean by "Observing a window of activity for the whole population is important for the model to predict behaviour in dynamics tasks". Also, very interesting to see your SWA results and indeed a bit bizarre that larger windows do not increase the performance.
> >
> > Even though I find your work very interesting and your findings quite important, I find that the added value of your work is solid, but not groundbreaking (I don't find the new scoring scheme of Neurips very helpful to be honest). In the old regime, I would give it an 8 but I wouldn't go to 9-10, since the main work even though important and valuable is mostly replacing the attention module with a recurrent model and engineering the model to run online.

---

> > > ### Author Response · Authors · 2025-08-05
> > >
> > > Thank you very much for your continued support, recognition of our contributions, and detailed response; we sincerely appreciate your comments.
> > >
> > > To clarify, we just meant that predicting behaviour per spike (i.e., one forward pass per spike) would be wasteful and difficult compared to making predictions after observing population activity every $T_C$ ms. With a quick attempt at replacing the input CA with a recurrent layer, using a buffer for the spikes, and running the model every 50ms, we were unable to achieve similar performance. However, we believe that this is a promising direction which requires some engineering: for example, figuring out the right input positional embeddings (since spike timing is irregular but important, and even though Mamba does not typically use PEs) or input chunking mechanism [1].
> > >
> > > 1. Hwang et al. "Dynamic Chunking for End-to-End Hierarchical Sequence Modeling." arXiv:2507.07955 (2025).
> > >
> > > Thank you again for your valuable suggestions and for engaging in discussion with us.

---

> > > > ### Author Response · Authors · 2025-08-05
> > > >
> > > > To supplement our comment above, here are some preliminary results with a fully recurrent encoder, where we use a GRU encoder instead of the input cross-attention (CA) and a GRU backbone to integrate information across time chunks. We maintain the same input configuration as with the input CA – contiguous, non-overlapping 50ms time chunks, and the encoder hidden state is reinitialised for each chunk (i.e., no state is maintained across chunks for the input-encoding GRU). We do not use any positional embeddings for the spikes (but they are ordered by spike-times) and our readout module remains the same (output CA + Linear). Results are shown in the table below.
> > > >
> > > > |Model|C–CO 2016|T–RT|
> > > > |-|-|-|
> > > > |POSSM-GRU-SS w/ recurrent encoder|0.9396 $\pm$ 0.0066|0.7334 $\pm$ 0.0611|
> > > > |POSSM-GRU-SS w/o spike times|0.9026 $\pm$ 0.0135|0.7207 $\pm$ 0.0632|
> > > > |POSSM-GRU-SS w/ spike times|0.9549 $\pm$ 0.0012|0.7687 $\pm$ 0.0669|
> > > >
> > > > Currently, only partial information about spike timing is provided to the model (through the temporal ordering of input spike tokens) due to the lack of positional embeddings – this could be one reason behind the performance gap. While we have not explored the entire design space with this configuration, we believe this experiment lays the foundation for important future directions we wish to pursue. We will endeavour to include these results in our Appendix.
> > > >
> > > > Thank you again for your time and insightful suggestions!

---

> ### Comment · Reviewer_Xwt5 · 2025-08-07
>
> Thank you for your detailed response and I am looking forward to see your final manuscript.

---

### Official Review · Reviewer_eRCh · 2025-06-29

**Clarity:** 4
**Significance:** 4
**Originality:** 3
**Rating:** 5
**Confidence:** 4

**Summary:**

In this paper, the authors propose a novel hybrid method, POSSM,  designed for generalizable, real-time neural decoding addressing limitations of traditional recurrent neural networks (RNNs) and transformer-based approaches. POSSM combines individual spike tokenization via a cross-attention module with a recurrent state-space model, enabling fast and causal online prediction on neural activity and efficient generalization to new sessions, individuals, and tasks through multi-dataset pre-training. It processes neural activity in 50ms chunks, tokenizing individual spikes.

The paper demonstrates that POSSM achieves comparable decoding accuracy with state-of-the-art Transformers at a fraction of the inference cost, making it suitable for real-time brain-computer interfaces as its inference time is well within the optimal latency. A notable contribution is its potential for cross-species transfer learning, shown by improved human handwriting decoding performance after pre-training on monkey motor-cortical recordings. POSSM also effectively decodes long-context human speech, a task computationally expensive for standard transformer models. These results position hybrid SSMs as a promising approach for practical, general neural interfaces.

**Questions:**

- Could the authors comment on how POSSM's applicability might be extended beyond motor areas? For example, how could it be adapted to cognitive or sensory domains where clear time-to-time correspondence is lacking? Addressing this would help establish its broader utility as a generalizable neural interface.
- Could the authors provide further insights into the interpretability of POSSM and how it may facilitates understanding of neural dynamics?

**Ethical Concerns:**

["NO or VERY MINOR ethics concerns only"]

**Final Justification:**

The paper introduces POSSM, a novel hybrid architecture for real-time neural decoding that successfully bridges the gap between accuracy, inference speed, and generalization. The demonstrated cross-species transfer learning, from non-human primates to human handwriting tasks, is particularly exciting and clinically relevant, addressing a major data acquisition challenge. The reported faster inference time compared to state-of-the-art Transformers is a critical advantage for real-time Brain-Computer Interfaces (BCIs).

The authors' rebuttal effectively addressed key concerns. Applicability beyond motor areas: The preliminary results decoding running speed from multi-region (visual cortex + subcortical) neural activity and the clarification on handling tasks without dense time-to-time correspondence significantly bolster the model's claimed generalizability.
Intrinsic Multimodality: New results showing improved performance when integrating binned spike counts and spike-band powers for speech decoding demonstrate the architecture's flexibility.

Remaining unresolved issues, such as the reliance on purely supervised training and in-depth interpretability, are reasonably acknowledged by the authors as important future work and not the primary focus of this paper. These limitations are also transparently discussed within the paper.

**Limitations:**

Yes.

**Paper Formatting Concerns:**

No.

**Quality:**

4

**Strengths And Weaknesses:**

Strengths
*   Superior decoding accuracy comparable to state-of-the-art transformers, yet it is significantly faster during inference. This speed is crucial for real-time BCIs, as its inference time is well within the optimal latency.
*   Efficient Generalization and Flexible Input Processing: Unlike traditional RNNs that struggle to generalize to new sessions or subjects due to rigid, fixed-size, time-binned inputs, POSSM leverages individual spike tokenization. This flexible input processing, combined with multi-dataset pre-training, allows POSSM to efficiently generalize to new sessions, individuals, and tasks.
*   Demonstrated Cross-Species Transfer Learning: A unique and impactful strength is POSSM’s capability for cross-species transfer learning. This provides a solution to the challenge of limited human data by leveraging a larger corpus of existing animal model data to enhance clinical outcomes.

Weaknesses
* Limited Evaluated Scope to Motor Areas: the tasks and data used for POSSM's evaluation are largely associated with the motor areas. Its performance and applicability outside of motor-related neural activity are not yet fully demonstrated.
* Reliance on Supervised Training: The current training approach for POSSM is supervised, meaning it requires labeled neural activity paired with behavioral or cognitive variables. This also pertains to the applicability outside of motor-related neural activity that might have no explicit behavioral labels.
* Lack of Intrinsic Multimodality: POSSM, in its current form, lacks inherent multimodality. While its architecture is flexible enough to accommodate different neural data modalities through proven tokenization schemes (e.g., Calcium Imaging, LFP), it is not intrinsically designed to seamlessly integrate and process multiple data types simultaneously, a potential area for future development.

---

> ### Author Rebuttal · Authors · 2025-07-30
>
> We thank you sincerely for your positive review of our paper and for outlining important research directions that we are actively investigating for future work. Below we respond to your questions and concerns:
>
> > W1. Limited Evaluated Scope to Motor Areas
>
> We acknowledge that in this work, we have mostly considered datasets with motor-cortical neurons and some motor-related behaviour. However, these datasets also contain neurons from Broca's area (speech task [1]) and S1 (primary somatosensory cortex; random target task [2]) – demonstrating that the method can process neurons from a broader set of neural regions.
>
> An investigation of POSSM's performance on data from other brain regions is an interesting direction for future work, and we have good reason to believe that it would work well beyond the motor domain. Notably, POYO+ [3], which adapts the POYO tokenization for regular time series data such as Calcium recordings (similarly to our tokenization for binned spike counts in the speech and handwriting experiments), demonstrates strong behavioural decoding performance on visual cortex recordings from the Allen Brain Observatory dataset [4]. As POSSM shares the same multitask readout framework and tokenization scheme as POYO while exhibiting similar performance, we expect that it will generalize well to other brain regions and tasks.
>
> To support this, we provide preliminary results on 5 random sessions from the Allen Visual Behavior Neuropixels dataset [4], where we decode running speed from multi-region neural activity (visual cortex + several subcortical areas including visual thalamic areas, hippocampus, and midbrain).
>
> |Model|Running Speed $R^2$|
> |-|-|
> |POSSM-SS-GRU|0.9295 $\pm$ 0.0326|
>
> This result provides early evidence that POSSM is effective across a variety of brain regions. We also anticipate that even better performance can be obtained with a mixture of more careful experimentation, training on multiple tasks [3], backbone changes, and hyperparameter tuning. Note that the sequence lengths at evaluation are 10x the training sequence lengths, again demonstrating strong length generalization.
>
> > W2. Reliance on Supervised Training
>
> We agree with the reviewer on the limitations of using a purely supervised learning paradigm. Indeed, requiring labeled data restricts both the amount and diversity of training data, potentially resulting in models that are biased towards certain tasks with denser or easier-to-curate lables. Moving away from supervised training and towards self-supervised learning (SSL) objectives is a key direction that we are investigating in ongoing work. With SSL, we aim to leverage unlabeled neural data to further improve downstream decoding performance, especially when finetuning to new sessions or subjects. We are also investigating the interpretability of the learned parameters, such as the unit embeddings [3], trained through SSL – specifically, whether we can identify finer details about the neural population (e.g., cell types, brain regions, etc.). Finally, we are also interested in how best to train an SSM using SSL objectives, compared to Transformer-based masked autoencoders [5]. As this is an ongoing effort, we leave discussion of our results for future work.
>
> > W3. Lack of Intrinsic Multimodality
>
> Thank you for mentioning multimodality, which is another important direction in our ongoing research. While we have only considered neural spiking data in this paper, we would like to point out that several of these datasets are recorded with multiple distinct "modalities", such as threshold crossings, sorted or unsorted spikes (a unit could be either a single neuron or a "multiunit" or "channel" depending on the dataset), and binned spike counts from public datasets [1,6]. To this end, we have recently trained POSSM on the speech task [1] **on both the binned spike counts and spike-band powers**. Excitingly, we observe strong performance when integrating both modalities (see table below).
>
>
> |Model|Validation PER ($\downarrow$)|
> |-|-|
> |Unidirectional GRU|21.74%|
> |Unidirectional POSSM-GRU|19.80% |
>
> Finally, in ongoing work, we are training our models on both spikes and local field potentials (LFPs) in order to investigate whether the same cross-attention layer can process both spikes and LFPs (with an additional modality embedding), or if other forms of early/late modality fusion work better. The tokenization scheme we use is identical in both these cases (LFPs are tokenized similarly to binned spike counts and Calcium imaging in POYO+ [3]), so we would argue that our approach is quite general and involves minimal preprocessing. We also thank you for acknowledging in your review that our architecture is "flexible enough to accommodate different neural data modalities through proven tokenization schemes".
>
> > Q1. Could the authors comment on how POSSM's applicability might be extended beyond motor areas? For example, how could it be adapted to cognitive or sensory domains where clear time-to-time correspondence is lacking? Addressing this would help establish its broader utility as a generalizable neural interface.
>
> This is an important question, and we have provided promising evidence (running speed decoding from Neuropixels data [4]) and clarifications (having neurons from S1 and Broca's area in our data mix) that POSSM extends beyond just motor areas in the response to W1. Furthermore, we already explore domains where time-to-time correspondence is lacking – the speech decoding task [1] does not have dense labels for each time chunk, so we use connectionist temporal classification (CTC) [7] to predict the best sentence response at the whole-trial level for the given neural activity sequence. We have also carried out an experiment on predicting cursor velocity 50ms or 100ms in the future, thus querying the output module with timestamps beyond the current time chunk. Results are provided in the table below. We believe that such flexibility will enable the use of POSSM in tasks such as those you have outlined.
>
> |Setup|C–CO 2016|T–RT|
> |-|-|-|
> |POSSM-GRU-SS 50ms future|0.9495 $\pm$ 0.0045|0.7498 $\pm$ 0.0874|
> |POSSM-GRU-SS 100ms future|0.9368 $\pm$ 0.0027|0.7252 $\pm$ 0.0813|
>
> > Q2. Could the authors provide further insights into the interpretability of POSSM and how it may facilitates understanding of neural dynamics?
>
> Thank you for the question. While the our work is focused on strong predictive performance and fast, real-time inference for neural decoding, we agree that interpreting how deep learning models arrive at their predictions is an important line of work.
>
> First, simple methods such as vanilla RNN/SSM models already help with understanding neural dynamics very well through dynamical systems analysis techniques, so we do not think POSSM can provide much additional benefit here. The spike tokenization scheme, on the other hand, can enable classification of brain regions and cell types [3] – the similarity of learned unit embeddings for neurons could potentially correlate with functional similarity of neurons across sessions or animals.
>
> As for the interpretability of POSSM itself, we looked at the evolution of hidden states from the POSSM-Mamba and POSSM-GRU models over test trials. Unfortunately, NeurIPS does not allow us to share figures or links for the rebuttal – nevertheless, we aim to describe our findings here. We found that the top 2 principal components (PCs) of the hidden states explained 68% and 51% of the variance for the Mamba and GRU respectively. For both models, the first PC resembled the magnitude of the cursor speed in upcoming time chunks, while the second PC seemed to represent change in cursor velocity (~acceleration, direction-sensitive). We leave more detailed analysis of the hidden states and various architectural components for future work. We would like to note, however, that enforcing learned latents to be human-interpretable rarely leads to improved downstream performance [8], while the objective of our work was to obtain strong performance under real-time inference constraints.
>
> ---
>
> References:
>
> 1. Willett et al. "A high-performance speech neuroprosthesis." Nature (2023).
> 2. O’Doherty et al. "Nonhuman primate reaching with multichannel sensorimotor cortex electrophysiology." Zenodo (2017).
> 3. Azabou et al. "Multi-session, multi-task neural decoding from distinct cell-types and brain regions." ICLR (2025).
> 4. de Vries et al. "Sharing neurophysiology data from the Allen Brain Observatory." Elife 12 (2023).
> 5. Ye et al. "Neural data transformer 2: multi-context pretraining for neural spiking activity." NeurIPS (2023).
> 6. Willett et al. "High-performance brain-to-text communication via handwriting." Nature (2021).
> 7. Graves et al. "Connectionist temporal classification: labelling unsegmented sequence data with recurrent neural networks." ICML 2006.
> 8. Mittal et al. "Does learning the right latent variables necessarily improve in-context learning?" ICML (2025).
>
> ---
>
> Once again, we thank you for your time and positive review, and hope that our response has addressed your concerns. We will endeavour to include all these additional results in our paper and supplementary material. We would be grateful if you would consider supporting our paper more strongly as a result. We are also happy to answer any other questions you may have and look forward to discussing further!

---

> > ### Comment · Reviewer_eRCh · 2025-08-02
> >
> > Thank you for the feedback, which has addressed most of my concerns. I believe this work will contribute to the community; therefore, I will maintain my original rating and recommend acceptance of the paper.

---

> > > ### Author Response · Authors · 2025-08-02
> > >
> > > Thank you very much for your positive evaluation of our work. If you have any outstanding concerns, please let us know and we'd be happy to address them. Again, thank you for your time!

---

### Official Review · Reviewer_Eajm · 2025-07-01

**Clarity:** 3
**Significance:** 2
**Originality:** 2
**Rating:** 5
**Confidence:** 4

**Summary:**

The paper introduces POSSM, an hybrid (SSM + Transformer) model to accelerate neural decoding inference time by leveraging the leaner SSM models while retaining a previously developed spike tokenization scheme. The new architecture blueprint allows for several SSM components instantiations (GRU, S4, Mamba). The authors have benchmarked the new method on monkey and human spike neural data recorded while performing motor tasks (plus a speech decoding task).

**Questions:**

- Is the gain of pre-training meaningful in a real-world setting, i.e. how many recording hours are needed for the single-session version to catch up with pretrained model?
- Val $R^2$ performance seems to be close between the single-session and the o-POSSM version, despite the latter being significantly bigger in size (8M vs <1M). Is scaling of the single-session to match the size of the o-POSSM version making an impact?
-  The o-POSSM has the same inference time as the POSSM-SS despite being ~x10 larger, how is this achieved?
- How does the particular tokenization scheme used compare with a more ViT-like (used by NDT-2 for example)? What are its benefits?

**Ethical Concerns:**

["NO or VERY MINOR ethics concerns only"]

**Final Justification:**

The authors have fully addressed my original concerns in their in-depth rebuttal, providing new results that showed: (1) that with multimodal features their model can indeed improve upon previously published results, (2) training with more data further improved their systems, (3) spike-times information during the tokenization scheme contributes significantly to the overall decoding performances, (4) all minor clarity issues/statements are addressed.

**Limitations:**

The authors have not provided an explicit section showcasing the limitations of the current technique.

**Paper Formatting Concerns:**

No major formatting concerns.

**Quality:**

2

**Strengths And Weaknesses:**

## Strengths
The authors have successfully combined previous SSM models with a spike-based tokenization scheme and develop a novel architecture which delivers similar performance while significantly reducing inference time, which is important for practical BMI applications. The paper is in general well written with clear figures.

## Weaknesses
- The overall evaluation suite report limited comparisons against alternative models. Beyond the similar POYO-1 model (which the current architecture is inspired to) Table 1 only reports comparisons against NDT-2 (with caveats on hyperparameters tuning). All other listed results represents variants of the proposed architecture.
- The claim that full fine-tuning yields better results than unit identification (Line 234-235) is fully expected.
- "[...] We see that pre-training results in a considerable initial boost in performance when fine-tuning to unseen data or new subjects." (Line 241-242). This claim seems to clash with what reported in either Figure 3a or 3b where o-POSSM and POSSM-SS have either comparable initial $R^2$ values as a function of training step (where o-POSSM starts actually lower, 3a) or as a function of number of samples (3b). Furthermore it seems that the gap seems to close after only ~32/64 trials, which sounds reasonable for calibrating a BCI.
- Unclear what data displayed in Figure 3c, right refers to. Probably results are from training on Monkey C-CO and testing on T-CO (based on comparing with Table 1), but the caption is unclear.
- The author evaluated their new family of models on the human speech decoding task, however they did not report any meaningful comparison for their results. As such it is very hard to assess whether the proposed model represents any improvements on the sota. In fact, the original publication (Willet et al. 2023) reported PER well below the authors own results (~20% vs. 27.32% for their best GRU model).
- Little to no details are provided for the o-POSSM variant which Figure 3c shows to be ~8M vs. <1M model, however it is unclear where this scaling takes place.

Typos & Minor:
- (Line 100) Ml --> ML
- (Line 125) retrain --> retain
- (Line 275-276) Text line width is incorrect

---

> ### Author Rebuttal · Authors · 2025-07-31
>
> We sincerely thank you for your detailed review. We address your comments below:
>
> > W1. The overall evaluation suite report limited comparisons ...
>
> In Table 1, we provide performance metrics for several non-POYO-style methods, including MLP, S4D, Mamba, GRU, and NDT-2, which can be used in a causal manner. These are popular, SOTA models for neural decoding from spikes [1], and unlike previous works, we have endeavoured to make these baseline models as performant as they can get. Notably, we have improved the results of the NDT-2 baseline through hyperparameter tuning and pretraining on all our data (see table). However, we demonstrate in Table 1 that POSSM handily outperforms all non-POYO baselines. If you have any additional suggestions on baseline methods, we would be happy to compare against them.
>
> |Model|C–CO 2010|T–CO|T–RT|
> |-|-|-|-|
> |NDT-2|0.7527 $\pm$ 0.0926|0.6897 $\pm$ 0.0657|0.6895 $\pm$ 0.0662|
> |Best non-POYO baseline (single session)|0.7308 $\pm$ 0.1089|0.8526 $\pm$ 0.0243|0.7279 $\pm$ 0.0679|
> |POSSM-SS best|0.7959 $\pm$ 0.0772|0.8863 $\pm$ 0.0222|0.7687 $\pm$ 0.0669|
> |o-POSSM-FT best|0.8216 $\pm$ 0.0945|0.9068 $\pm$ 0.0170|0.7741 $\pm$ 0.0683|
>
> > W2. The claim that full fine-tuning yields better results ...
>
> We would like to clarify that this statement should not be seen as a new claim, but as an interpretation of our results. We are emphasizing that - especially on new subjects - full finetuning (FT) has better performance overall, although it remains notable that unit identification (UI) is close to or better than single session performance (see next response). We will rephrase this statement to clearly indicate this. Again, we would like to kindly point out that this is not our main claim or contribution.
>
> > W3. This claim seems to clash with what reported in either ...
>
> The initial results at training step 0 are random because unit and session embeddings are randomly initialized. Thus, the variance in performance at the initial steps is expected to be high – we do have runs where o-POSSM starts higher than POSSM-SS. We would like to emphasize that the key takeaway of Fig. 3a is that UI leads to much faster convergence and higher performance than single session models (see table below), in less than 1/3rd of the training steps and by finetuning far fewer parameters (~10K vs ~600K).
>
> |Dataset|Best POSSM-SS|Best o-POSSM-UI|Best o-POSSM-FT|o-POSSM vs. POSSM improvement|
> |-|-|-|-|-|
> |C–CO 2010|0.7959 $\pm$ 0.0772|0.8022 $\pm$ 0.0818|0.8216 $\pm$ 0.0945|3.23%|
> |T–CO|0.8863 $\pm$ 0.0222|0.8921 $\pm$ 0.0174|0.9068 $\pm$ 0.0170|2.31%|
> |T–RT|0.7687 $\pm$ 0.0669|0.7464 $\pm$ 0.0692|0.7741 $\pm$ 0.0683|0.70%|
> |NLB Area2|0.8350|0.8394|0.8941|7.08%|
>
> These improved metrics are a result of pretraining on more data, now including all sessions from the Churchland et al. dataset [2]. With this, we have improved the performance of our few-shot finetuning as well:
>
> |Model/Trials|4|8|16|32|64|128|
> |-|-|-|-|-|-|-|
> |POSSM-SS|0.4607 $\pm$ 0.0474|0.6010 $\pm$ 0.0997|0.8390 $\pm$ 0.0375|0.8929 $\pm$ 0.0267|0.9410 $\pm$ 0.0057|0.9463 $\pm$ 0.0029|
> |o-POSSM-FT|0.5330 $\pm$ 0.1374|0.7410 $\pm$ 0.1097|0.9122 $\pm$ 0.0163|0.9410 $\pm$ 0.0055|0.9508 $\pm$ 0.004|0.9582 $\pm$ 0.001|
>
> |Model/Trials|4|8|16|32|64|
> |-|-|-|-|-|-|
> |POSSM-SS|0.2979 $\pm$ 0.0324|0.3892 $\pm$ 0.0152|0.5291 $\pm$ 0.0097|0.6523 $\pm$ 0.0121|0.7064 $\pm$ 0.0138|
> |o-POSSM-FT|0.3613 $\pm$ 0.0628|0.5372 $\pm$ 0.0103|0.6157 $\pm$ 0.0143|0.6709 $\pm$ 0.0268|0.7087 $\pm$ 0.0148|
>
> The performance improvements from pretraining are now more apparent both for fewer and more trials. Furthermore, we would argue that in the "real world", calibration should be as data- and compute-efficient as possible, so the fewer trials needed to calibrate the BCI, the better (64 trials = >5 minutes of active task performance, which can get cumbersome if done everyday).
>
> > W4. Unclear what data displayed in Figure 3c ... training on Monkey C-CO and testing on T-CO ...
>
> Yes, you are correct – we will clarify this in the caption. Thank you for pointing out this clarity issue. While NeurIPS does not allow us to provide figures with the rebuttal, we have updated the caption to be clearer and modified all relevant figures to indicate the subjects and sessions clearly in our draft manuscript.
>
> > W5. The author evaluated their new family of models on the human speech decoding task ...
>
> In our paper, we initially considered the problem of decoding speech from neural spikes alone. Importantly, the GRU you are referring to is achieved with spike-band powers included (we reproduced PER = 21.74%). This is a **multimodal** approach that is outside the scope of this paper, but we are actively exploring such approaches in ongoing work. As a preliminary experiment, we provide performance comparisons between the baseline and POSSM. We achieve competitive performance, even before full exploration of the design space. We believe with an improved tokenization scheme and access to more data, a significant bump in performance can be attained.
>
> |Model|Val PER ($\downarrow$)|
> |-|-|
> |Unidirectional GRU|21.74%|
> |Unidirectional POSSM-GRU|19.80%|
>
> Our work focuses on PER, not downstream language modeling, which many prior works optimize via ensembling and bidirectional GRUs (most other methods such as transformers and SSMs do not outperform the GRU, and most top teams use biGRU decoders with training tricks [1]). These methods also lack public code and do not report PER scores [1], making direct comparisons difficult. In contrast, our unidirectional model does not have these speech-specific optimizations and is designed to be general-purpose, yet still attains competitive performance. While the current dataset is small, we expect our latent representations to become more expressive when trained with larger datasets, such as the upcoming Brain2Text-2025 benchmark. We are also exploring finetuning a model pre-trained on motor tasks, similar to what we did for handwriting.
>
> > W6. Little to no details ...
>
> Apologies that this was not clear. Currently, these details are provided in the Appendix (Section S.5.2, Table 5), but we have now updated our draft to include these details in the main paper. Scaling is primarily determined by the number of SSM/RNN layers, and also by an increase in the dimensionality of the input and RNN hidden state.
>
> > Q1. Is the gain of pre-training meaningful ...
>
> This proposed study is made difficult by the fact that sessions in existing datasets have limited amounts of data. However, we show that single-session training is far more expensive, as it requires training ~600K params while UI only needs to tune ~10K. While being parameter-efficient, we also know that UI achieves higher performance more quickly than training from scratch. These benefits are especially felt when there are multiple downstream subjects to whom the model must be finetuned, as the reduced training times and parameter counts are compounded. Finally, UI outperforms single-session especially in low data regimes (c.f. finetuning results above), which are reflective of BCI use cases, indicating that UI is a viable and useful approach in real-world settings.
>
> > Q2. Val $R^2$ performance seems to be close between ...
>
> In our updated results shared above, we show gains from additional pretraining for o-POSSM. We decided to scale up the size of the model only if there was a performance benefit from doing so, and we believe that with even more pretraining data, the gains would be higher. We would also like to note that the variance in sizes for o-POSSM variants is quite high (S4D 4.5M, GRU 7.9M, Mamba 9M).
>
> > Q3. The o-POSSM has the same inference time ...
>
> The o-POSSM models are slightly slower on average, especially on CPU. The similarity in inference times is simply the result of PyTorch having efficient, parallelizable implementations of various architectural components (attention and GRU), we do nothing in addition to ensure the closeness in inference times. We are, however, working on using Flash Attention for our initial cross-attention layer to further speed up computations, especially during training.
>
> > Q4. How does the particular tokenization scheme ...
>
> The ViT-like patching scheme used in NDT-2 groups together arbitrary neurons within a patch, does not have a mechanism to identify separate units, and is not permutation-invariant like our scheme. Meanwhile, we explicitly learn embeddings for each unit, and we believe that this scheme both generalizes better than and enables a more straightforward and efficient finetuning procedure than NDT-2 by relearning just the unit/session embeddings.
>
> Furthermore, we believe that using spike timing information could be instrumental in improving decoding performance. In the table below, we show results with and without access to raw spike times for each spike, where the latter would be identical to binning-based approaches. We see a considerable performance drop in the latter case, thus further motivating our choice of tokenization scheme.
>
> |Dataset|POSSM-Mamba-SS w/ spike times|w/o spike times|
> |-|-|-|
> |C–CO 2010|0.7959 $\pm$ 0.0772|0.7040 $\pm$ 0.0903|
> |T–RT|0.7418 $\pm$ 0.0790|0.7275 $\pm$ 0.0734|
> |NLB Area2|0.8350|0.5568|
>
> We believe that these differences in tokenization could be the key factor behind the gap in performance between POSSM/POYO-style models and NDT-2. In ongoing work, we are also investigating how well the unit embeddings encode neuronal properties when trained with SSL objectives.
>
> References:
>
> 1. Willett et al. "Brain-to-Text Benchmark '24: Lessons Learned." arXiv:2412.17227 (2024)
> 2. Churchland et al. "Neural population dynamics during reaching." Nature (2012).
>
> We thank you once again for your review and attention to detail, and believe your comments have helped improve our work. We hope we have addressed your concerns, and if so, we kindly ask if you would be willing to reconsider your score. We look forward to discussing further with you!

---

> > ### Comment · Reviewer_Eajm · 2025-08-02
> >
> > I would like to thank the authors for their in-depth rebuttal which addressed all of my questions and provided new interesting results. I found particularly relevant the results obtained with more data (W3) and the speech decoding results (W5) when using *multimodal* features (apologies for having originally misunderstood this important difference). I sincerely believe that these new results are valuable additions which strengthen the original submission.
> >
> > Another particularly interesting new insight is the result for the comparison of different tokenization schemes (Q4) which shows that exact spike-times can indeed boost performance (sometimes quite dramatically as in the case of the NLB Area2 dataset!). I believe this result is yet another valuable addition which not only helps explaining why POYO-style models enjoys good performances, but will in general inform the whole field in designing future architectures.
> >
> > I thank again the authors for their excellent rebuttal efforts and I will modify my score accordingly to recommend the manuscript for acceptance.

---

> > > ### Author Response · Authors · 2025-08-02
> > >
> > > Thank you very much for your response! We will include these new results clearly in our final manuscript. We sincerely thank you for your time and for your questions that have enriched our submission.

---

### Official Review · Reviewer_oDCz · 2025-07-01

**Clarity:** 3
**Significance:** 3
**Originality:** 2
**Rating:** 4
**Confidence:** 5

**Summary:**

Paper: Generalizable, real-time neural decoding with hybrid state-space models.
The paper addresses an unmet need in neural decoding: how to maintain fast, real-time causal inference while supporting generalization across subjects, sessions, and tasks. This is a longstanding challenge in closed-loop BCIs, where models have to adapt to new neural distributions without retraining from scratch, and where computational latency has to remain within strict limits (<10ms). Most existing models either generalize poorly or are too slow for real-time deployment. However, the improvements are marginal.

**Questions:**

see above

**Ethical Concerns:**

["NO or VERY MINOR ethics concerns only"]

**Final Justification:**

My concerns have been addressed and I revised up my evaluation.

**Limitations:**

see above

**Paper Formatting Concerns:**

generally clear

**Quality:**

3

**Strengths And Weaknesses:**

Novelty of methods/approach
The state-of-the-art for neural decoders rely on either binned spike counts with RNNs or long-context Transformers, both of which struggle with either generalization or real-time constraints. This paper introduces a hybrid architecture that processes individual spikes as tokens, rather than binned aggregates, and compresses them into a latent representation suitable for efficient recurrent modeling. What’s new is the combination of spike-level tokenization with chunk-wise cross-attention and a recurrent backbone, enabling both fast, causal inference and generalization across tasks, sessions, and even species.


Significance of results
POSSM matches or exceeds performance of state-of-the-art models across three domains: monkey reaching, human handwriting, and human speech decoding. Of particular note is the successful transfer of dynamics from pretraining on NHP datasets to a human subject handwriting task—an important result for neuroprosthetic applications where human data is limited. The architecture also achieves 5–9× faster inference than transformer baselines, including on CPU, making it suitable for embedded or clinical BCI systems.

Clarity of presentation
The paper is clearly structured and easy to follow. The figures effectively convey the model architecture and benchmarking results. The experimental comparisons are extensive and well-labeled. One minor suggestion would be to include a small schematic highlighting differences in causal vs. non-causal evaluation modes.

Technical correctness
​​The methodological details are rigorous and complete. The authors compare against appropriate baselines, including both from-scratch and pretrained models, and provide detailed breakdowns of performance across subjects, tasks, and fine-tuning strategies. The use of causal decoding and inference benchmarks suggests the claims are well-supported.

---

> ### Author Rebuttal · Authors · 2025-07-31
>
> We thank you for recognizing the several significant contributions of our work, including our fast, real-time inference, generalization capabilities, and successful transfer of dynamics from NHP datasets to a human handwriting task. Below, we address the points from your review:
>
> > However, the improvements are marginal.
>
> We would like to respectfully note that this point appears to be in contradiction with other points in the review, e.g., "POSSM matches or exceeds performance of state-of-the-art models across three domains...", "Of particular note is the successful transfer of dynamics from pretraining on NHP datasets to a human subject handwriting task—an important result for neuroprosthetic applications where human data is limited", and "The architecture also achieves 5–9× faster inference than transformer baselines...".
>
> First, we would like to emphasize that the primary goal of our work is not just to improve decoding performance, but to do so in a way that maintains strong performance **while generalizing easily to new subjects and being efficient for real-time use**. We have also endeavoured to make our baselines as competitive as possible. Nevertheless, we have demonstrated that our method consistently matches or outperforms several popular and state-of-the-art baselines. In new experiments, we also pretrained our large-scale, multi-session o-POSSM models on an expanded dataset that now includes all 10 sessions from the Churchland et al. [1] dataset. With this, we significantly improved the performance of our unit identification finetuning to be comparable to or better than single-session performance, while maintaining or improving full finetuning performance. We provide these results in the table below.
>
> |Dataset|Best POSSM-SS|Best o-POSSM-UI|Best o-POSSM-FT|o-POSSM vs. POSSM improvement|
> |-|-|-|-|-|
> |C–CO 2016|0.9550 $\pm$ 0.0003|0.9607 $\pm$ 0.0036|0.9615 $\pm$ 0.0056|0.68%|
> |C–CO 2010|0.7959 $\pm$ 0.0772|0.8022 $\pm$ 0.0818|0.8216 $\pm$ 0.0945|3.23%|
> |T–CO|0.8863 $\pm$ 0.0222|0.8921 $\pm$ 0.0174|0.9068 $\pm$ 0.0170|2.31%|
> |T–RT|0.7687 $\pm$ 0.0669|0.7464 $\pm$ 0.0692|0.7741 $\pm$ 0.0683|0.70%|
> |NLB Area2|0.8350|0.8394|0.8941|7.08%|
>
> With these new models, we have further improved our results on data efficiency (few-shot finetuning). We provide these results in the tables below, on a CO session:
>
> |Model|4 Trials|8 Trials|16 Trials|32 Trials|64 Trials|128 Trials|
> |-|-|-|-|-|-|-|
> |POSSM-SS|0.4607 $\pm$ 0.0474|0.6010 $\pm$ 0.0997|0.8390 $\pm$ 0.0375|0.8929 $\pm$ 0.0267|0.9410 $\pm$ 0.0057|0.9463 $\pm$ 0.0029|
> |o-POSSM-FT|0.5330 $\pm$ 0.1374|0.7410 $\pm$ 0.1097|0.9122 $\pm$ 0.0163|0.9410 $\pm$ 0.0055|0.9508 $\pm$ 0.004|0.9582 $\pm$ 0.001|
>
> and an RT session:
>
> |Model|4 Trials|8 Trials|16 Trials|32 Trials|64 Trials|
> |-|-|-|-|-|-|
> |POSSM-SS|0.2979 $\pm$ 0.0324|0.3892 $\pm$ 0.0152|0.5291 $\pm$ 0.0097|0.6523 $\pm$ 0.0121|0.7064 $\pm$ 0.0138|
> |o-POSSM-FT|0.3613 $\pm$ 0.0628|0.5372 $\pm$ 0.0103|0.6157 $\pm$ 0.0143|0.6709 $\pm$ 0.0268|0.7087 $\pm$ 0.0148|
>
> Finally, we have also integrated an additional modality – spike-band powers – in our speech decoding model, allowing us to further improve the performance over our previous results and the highly performant GRU baseline [2].
>
> |Model|Validation PER ($\downarrow$)|
> |-|-|
> |Unidirectional GRU|21.74%|
> |Unidirectional POSSM-GRU|19.80%|
>
> Overall, we believe that our results not only perform comparably with or advance the state-of-the-art on several tasks, but also maintain efficiency both in terms of finetuning data and inference times. If you have any other questions or concerns about our results, we would be happy to address them during the discussion phase!
>
> > One minor suggestion would be to include a small schematic highlighting differences in causal vs. non-causal evaluation modes.
>
> Thank you for the suggestion! While NeurIPS does not allow us to add figures or links in the rebuttal, we will include these in our final paper.
>
> ---
>
> References:
>
> 1. Churchland et al. "Neural population dynamics during reaching." Nature (2012).
> 2. Willett et al. "Brain-to-Text Benchmark'24: Lessons Learned." arXiv:2412.17227 (2024).
>
> ---
>
> Overall, while we are thankful for your positive comments, we are quite confused by the "borderline reject" rating of our paper as the review does not list specific weaknesses or questions for us to address. We would be grateful for any clarifications you might be willing to offer regarding your assessment, as this would help us better understand the recommendation. If our responses here and to other reviewers have addressed your concerns, we kindly ask if you would be willing to reconsider your score. We are also happy to address any concerns and questions you might have during the discussion phase, and eagerly look forward to your response.

---

> > ### Comment · Reviewer_oDCz · 2025-08-04
> >
> > I thank the authors for their discussion. I have revised my review up slightly as a result.

---

> > > ### Author Response · Authors · 2025-08-04
> > >
> > > Thank you for acknowledging our rebuttal and revising your assessment! If you have any outstanding questions, we are happy to address them. Thank you again for your time.

---

### Decision · Program_Chairs · 2025-09-17

**Decision:**

Accept (poster)

**Comment:**

This work introduces POSSM, a brain-computer interface (BCI) model that uses a hybrid transformer-state space model architecture to achieve state-of-the-art decoding performance in real time. POSSM uses spike-based tokenization (à la POYO), input cross-attention, and an SSM recurrent backbone to produce embeddings that allow for accurate decoding while lowering model inference times. POSSM is then applied to a variety of benchmark tasks (monkey reach, human handwriting, speech decoding) with state of the art results.

Reviewers were enthusiastic about the novel architecture, rigorous benchmarking, and careful experiments that show the value of more data and pretraining in generalizing across sessions and participants. Reviewers noted some small weaknesses in performance improvements and breadth of model comparisons, which the authors addressed during rebuttal with new results showing improved performance with additional data.

In all, a strong contribution to the literature on large-scale brain decoding models.